# Sampling errors and variability in video transects for assessment of reef fish assemblage structure and diversity

Stijn Bruneel[1,2]*, Long Ho[1], Wout Van Echelpoel[1], Amber Schoeters[1], Heleen Raat[1], Tom Moens[2], Rafael Bermudez[3,4], Stijn Luca[5], Peter Goethals[1]

**1** Department of Animal Sciences and Aquatic Ecology, Ghent University, Ghent, Belgium, **2** Marine Biology Research Group, Ghent University, Ghent, Belgium, **3** Galapagos Marine Research and Exploration, GMaRE, Joint ESPOL-CDF Program, Charles Darwin Research Station, Galapagos Islands, Ecuador, **4** Facultad de Ingeniería Marítima y Ciencias del Mar, Escuela Superior Politécnica del Litoral (ESPOL), Campus Gustavo Galindo, Guayaquil, Ecuador, **5** Department of Data Analysis and Mathematical Modelling, Ghent University, Ghent, Belgium

* stijn.bruneel@ugent.be

**Data Availability Statement:** The data, code and documentation can be found here: https://doi.org/

## Abstract

Video monitoring is a rapidly evolving tool in aquatic ecological research because of its non-destructive ability to assess fish assemblages. Nevertheless, methodological considerations of video monitoring techniques are often overlooked, especially in more complex sampling designs, causing inefficient data collection, processing, and interpretation. In this study, we discuss how video transect sampling designs could be assessed and how the inter-observer variability, design errors and sampling variability should be quantified and accounted for. The study took place in the coastal areas of the Galapagos archipelago and consisted of a hierarchical repeated-observations sampling design with multiple observers. Although observer bias was negligible for the assessment of fish assemblage structure, diversity and counts of individual species, sampling variability caused by simple counting/detection errors, observer effects and instantaneous fish displacement was often important. Especially for the counts of individual species, sampling variability most often exceeded the variability of the transects and sites. An extensive part of the variability in the fish assemblage structure was explained by the different transects (13%), suggesting that a sufficiently high number of transects is required to account for the within-location variability. Longer transect lengths allowed a better representation of the fish assemblages as sampling variability decreased by 33% if transect length was increased from 10 to 50 meters. However, to increase precision, including more repeats was typically more efficient than using longer transect lengths. The results confirm the suitability of the technique to study reef fish assemblages, but also highlight the importance of a sound methodological assessment since different biological responses and sampling designs are associated with different levels of sampling variability, precision and ecological relevance. Therefore, besides the direct usefulness of the results, the procedures to establish them may be just as valuable for researchers aiming to optimize their own sampling technique and design.

10.5281/zenodo.6759858 DOI: 10.5281/zenodo.6759858.

**Funding:** The research was partly funded by VLIR-UOS Biodiversity Network Ecuador, Special Research Fund of UGent and VLIR-UOS Global Minds. Travelling scholarships for staff and students were provided by FWO, CWO (UGent), VLIR-UOS (Global Minds) and VLIR-UOS Biodiversity Network Ecuador. The funders had no role in study design, data collection and analysis, decision to publish, or preparation of the manuscript.

**Competing interests:** The authors have declared that no competing interests exist.

# 1 Introduction

Anthropogenic pressures are known to affect the physical habitat and chemical water conditions of many reef ecosystems [1]. Although biotic indicators, such as the diversity and structure of reef fish assemblages, have proven useful to quantify environmental and ecological changes [2–4], the conclusiveness of any ecological or behavioral field study heavily depends on a proper choice of the sampling method and sampling design. When assessing the abundance, diversity and species composition of fish assemblages, visual census has been the preferred non-destructive method, with stationary point counts and strip transects being the most common options [5–7]. As such methods require substantial training and considerable time in the field, the number of locations that can be sampled is limited [8]. In addition, such traditional visual census techniques are often characterized by high detection heterogeneity, caused by observer bias (Table 1) and observer effects (Table 1) [9–15]. Observer effects are often considered as the spatiotemporal variation in observer bias due to gained experience or replacement of observers [8]. We argue, however, that the proceeding of sampling through time and space will not only alter the perception of the environment, but also the environment itself. More specifically, observers might attract or deter species [16, 17], while also gaining more experience and an altered perception [18]. Therefore, although partly related, observer bias is considered the result of consistent differences in the perception of a unique set of observers, while observer effects are considered the result of differences in observers' perception through time and space in combination with fish species-specific traits such as detectability (Table 1) and response to presence of observers. These observer effects in combination with random counting/detection errors and "random" or instantaneous fish displacement (see Table 1 and S1 File) at very short time scales (e.g. minutes) constitute the sampling variability which is referred to as baseline or instantaneous variability in observational fish assemblage studies [19, 20]. This sampling variability should be accounted for (or at least acknowledged) when designing sampling schemes and interpreting data [19, 20].

Nowadays, stationary video cameras are often used to remove observer bias, to reduce the impact on fish behavior and to be able to store the video data for future studies and validation [21–23]. Nevertheless, stationary cameras often only provide a limited view of the study area as only small patches are monitored, and are therefore of limited use when a rapid, yet spatially extensive survey of a study area has to be done [14]. Baited stationary cameras remediate this limitation by attracting nearby fish thereby providing a more representative assessment of the local fish assemblages [24]. However, due to the point-based nature of these camera-systems, it remains difficult to accurately account for habitat heterogeneity and assess habitat preferences (especially of less mobile species) [14, 25]. Because fish species, and in particular reef fish, are often strongly associated with specific micro-habitats [26], a more mobile approach may often be more appropriate for more cost-effective monitoring of fish densities [27]. Video transects

**Table 1. Glossary.**

| | |
|---|---|
| Design error | Error associated with the methodological characteristics of the sampling technique (e.g. transect length, swimming speed) |
| Detectability | Probability of observing a particular species during a given sampling occasion conditional on its presence at that location [16, 47] |
| Instantaneous variability | Sampling variability caused by the instantaneous displacement of fish, observer effects and random counting/detection errors [19, 20] |
| Observer bias | Effect of consistent differences in the perception of a unique set of observers on fish count |
| Observer effects | The effect of differences in observers' perception through time and space in combination with fish species-specific traits such as detectability and response to presence of observers on fish count |

still suffer from observer bias and observer effects, but the possibility to store video data for later analysis, to reduce the amount of time spent on field work and to standardize observations, are important incentives to upgrade traditional visual census techniques [28, 29]. Video transects have already been applied in numerous studies [27, 30], and remotely operated vehicles (ROVs) are likely to soon replace camera-operating humans in a variety of applications [31, 32]. Although studies comparing different video and other visual census techniques are becoming more frequent [10, 25, 29, 33–35], few have focused on the methodological aspects of video transects, such as observer bias (Table 1), design errors (Table 1), sampling variability, counting metrics, data types and data transformations [36, 37]. Video transects are often considered a simple extension of visual census transects, although the ability to store and standardize video observations (see S2 File) justifies a specific methodological assessment of the technique. The aim of this study was therefore to provide a detailed assessment of the aforementioned methodological aspects and overall suitability of video transects to study reef fish assemblages.

Since the results of methodological studies are meant to steer the methodology used in applied studies, it is crucial to understand and limit avoidable differences between both types of studies. On the one hand, applied ecological studies aiming to understand ecological processes and how they are affected by management practices typically make use of multiple random transects within one location to obtain a general picture of the local fish assemblage, each transect being covered only once by a single observer [6, 38, 39]. The lack of repeated observations renders any assessment of the variability related to observers, transects and methodological parameters difficult [40]. On the other hand, applied studies are often characterized by complex hierarchical designs to account for ecological gradients, while methodological studies typically entail more simple sampling designs [9, 33, 41], introducing a conceptual mismatch between both. Besides differences in the sampling design, there are often also differences in the statistical tools used by both types of studies. While applied studies typically involve either multivariate (for fish assemblages) and/or univariate (for single species) models [6, 25, 42, 43], methodological studies generally apply the latter [8, 10, 44, 45], but there are exceptions which integrate both [11, 22, 46].

To ensure a close connection between methodology and application while maintaining a sound statistical basis to assess the importance of different factors, the present study laid out a hierarchical design of repeated fixed video transects, with each transect being covered multiple times by multiple observers. Although repetitions of transects are not common in literature, they were necessary here to provide answers to the following research questions, which are of interest for both non-repetitive and repetitive studies. The innovative, yet complex design of this study can provide new insights in the methodology of observation techniques, yet requires multiple advanced statistical methods. An overview of these methods and reasoning behind their use is provided in section 2.2. The following research questions were addressed for both multivariate (for fish assemblages) and univariate (for single species) variables of interest.

- Are there (species-specific) observer effects which cause dependence between observations?

- How important are observer bias and sampling variability?

- How important are the methodological parameters transect length, number of repeats, counting metric and data type/ transformation?

## 2 Materials and methods

### 2.1 Data collection

The study took place in the Galapagos archipelago as part of an extensive project to map and monitor the unique, yet fragile, coastal ecosystems of the islands. At ten locations, fixed

transects were used to assess reef fish assemblages in rocky habitats close to the coast. The locations (n = 10) were situated along the coasts of two cities of two different islands; Puerto Ayora (Academy Bay) on Santa Cruz island (n = 5) and Puerto Velazco Ibarra on Floreana island (n = 5) (S1 Fig). In each location, 3 transects with a length of 50 meters each were laid out using ropes. For fish to be included they had be recorded within 2.5 m of either side of the transect line (area = 50 x 5 = 250 m$^2$). Observers and image analysts were trained to recognize whether fish had to be considered, depending on the estimated distance from the rope. It should be noted that, despite their training, observers are known to over or underestimate the distance from the rope, which will affect the inter-observer variability and/or sampling variability to some extent [36, 48]. All transects were monitored at a constant depth of 1.5 meter, parallel to the coastline, and only locations with a limited exposure to waves were selected to guarantee the safety of the observers and to avoid predominant effects of wave exposure on the structure of fish assemblages. All transects were approximately 20 meters apart. To quantify observer bias and sampling variability, a sufficiently high number of repeats was required, yet the maximal number of observations per day was limited by light availability and fatigue of the observers. Balancing these requirements resulted in each of the transects being recorded six consecutive times with single GoPro cameras (GoPro Hero 5 Black, 1080p, 60fps, wide FOV) by three different observers equipped with a mask and snorkel. Hence, 2 islands x 5 locations x 3 transects x 3 observers x 6 repeats yielded 540 observations (Fig 1).

The observers covered the transects in a browsing fashion, similar to the widely used S-type transects introduced by [28]. Observers browse through the transect at a fixed speed, but varying angle and can zoom in if needed to find individuals hiding in crevices. This S-transect was chosen over the more standarized I-transect with fixed angle and without zooming to enable detection of more individuals and species [27, 28]. Transects were placed during low tide and monitoring occurred during flood tides of consecutive days from the 19$^{th}$ until the 31$^{st}$ of August 2017. Per day, one site was surveyed. The duration of each observation was approximately 4.6 minutes, but varied (±0.7 minutes). The time between successive observations was at least one minute. The duration between observations was most often too short to consider observations as independent [49]. These dependencies between observations are described as observer effects, which will be discussed further. Observers covered the transects back to back, so three times in one direction and three times in the other direction. The analysis of the videos included species identification and the estimation of the number of individuals per species. Each fish identification and count was labeled with a time stamp to allow later splitting of the data in different subsets. Fish were counted in such a way that both MaxCount (total number of individuals per species) and MinCount (maximum number of individuals per species in one frame, also referred to as MaxN: to avoid double counting) could be determined (see section S3 File). Since MaxCount was found most appropriate for our video transects (see section 3.4), which is in line with results from previous studies on video transects [27, 30], it was used instead of MinCount. The videos were randomly assigned to one of two video analysts to reduce any potential experience bias from improving identification skills [18]. The video analysts had a similar level of experience. Each video was only analyzed once. Each observation was split up in 5 distance intervals (0–50, 5–45, 10–40, 15–35, 20–30 meters), based on the time stamps of the fish identifications, to assess the added value of increasing transect length. The exact distance could not be determined, but the observers aimed to maintain a constant swimming speed, so distance could be estimated from the time stamps. Although the inability to provide exact distances will introduce some additional variability that cannot be quantified, we assumed this variability to be stochastic and limited as observers were specifically trained to maintain a constant swimming speed. Because we aimed to provide a quick and mobile assessment of the local fish assemblages and limit the sampling effort per site, it was decided to

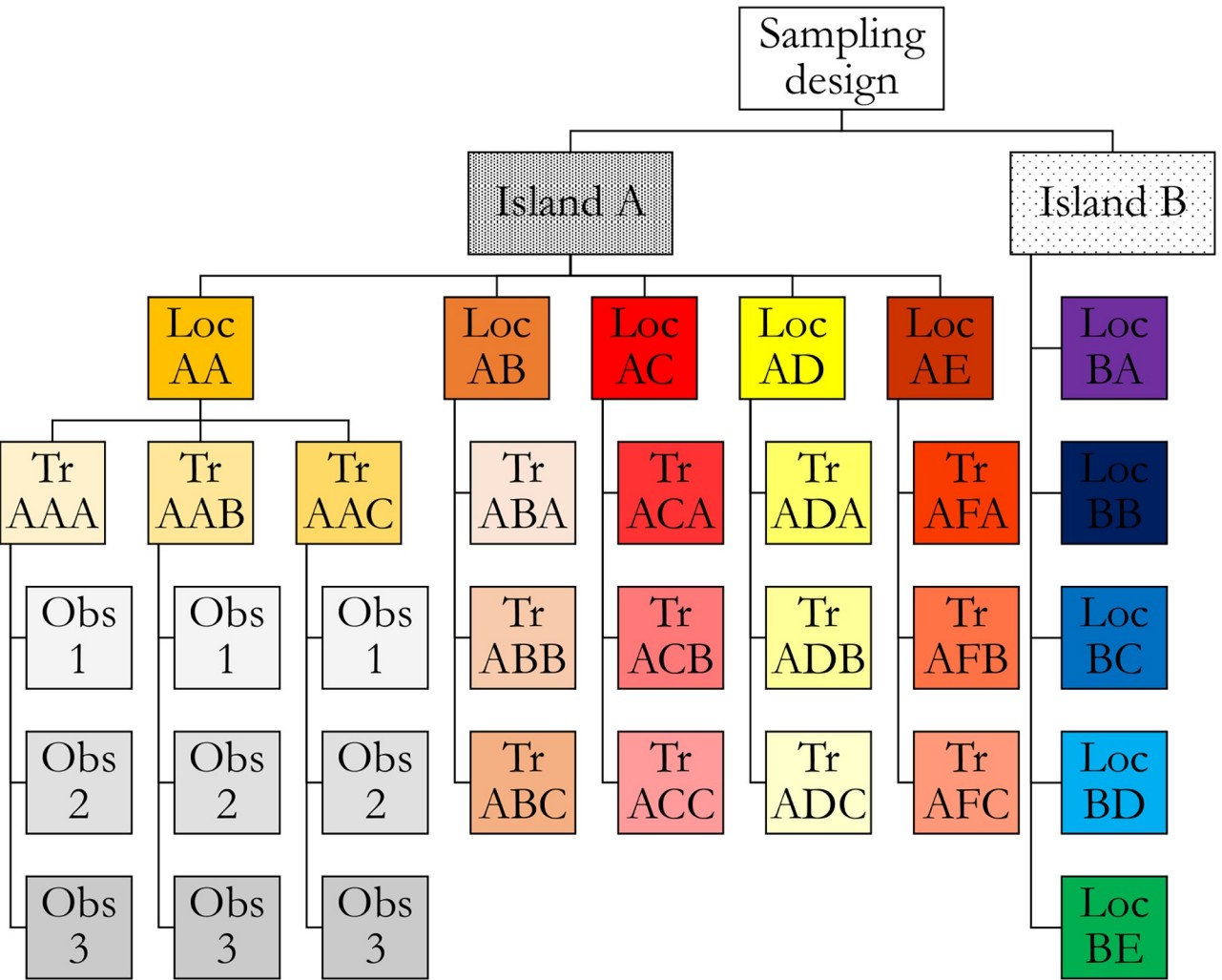

**Fig 1. Sampling design of the study.** Three spatial nested levels can be identified: Island, Location (Loc) and Transect (Tr). Each observer (Obs) covered each transect six times. For clarity reasons, not all subdivisions are presented in this figure. All transects of the different locations of island A were covered by the three observers and the locations of island B were further subdivided in transects, each covered by all three observers. The sampling design was balanced.

count all fish species [34, 36] and not to count discrete groups of species in transects of different, earlier established, optimal designs [19, 35]. All methods were carried out in accordance with the relevant guidelines and regulations of the Galapagos National Park Directorate under research permit PC-02–19. All experimental protocols were reviewed and approved by the Galapagos National Park Directorate Applied Research Department, which assesses animal care in research activities.

## 2.2 Data analysis

To understand patterns in the data, models for variation partitioning were developed. For this models to be relevant for methodological assessments, it is important to translate model results back to the context of data collection and sampling design (Fig 2). Although ideally the inter-observer variability of our models would only contain the observer bias, the potential temporal dependence of observations over observers might cause the estimates to be inflated by the

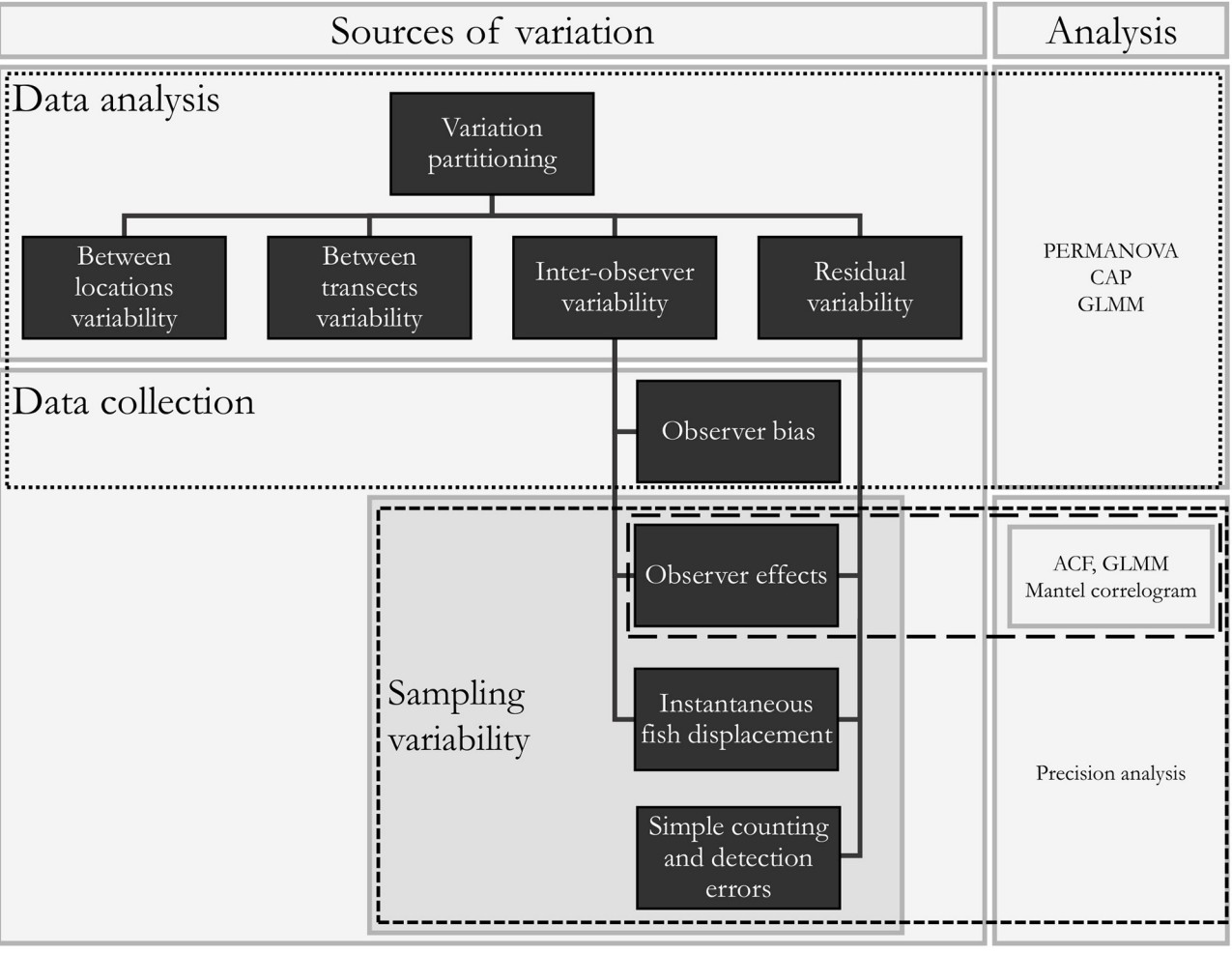

**Fig 2. Methodological framework of the study.** Variation partitioning using PERMANOVA (permutational multivariate analysis of variance), CAP (canonical analysis of principal components), and GLMM (generalized linear mixed models) to quantify the different sources of variation for multivariate and univariate data respectively. The modeled inter-observer variability not only comprises the observer bias, but also sampling variability, i.e. observer effects and instantaneous fish displacement. The residual variability comprises the remaining sampling variability. ACF (auto-correlation functions), GLMM and partial Mantel correlograms to assess observer effects. Precision analysis was used to assess the effect of number of repeats and transect length on quantification and reduction of sampling variability.

variability caused by observer effects and instantaneous fish displacement. Indeed, because observers covered transects sequentially the disturbances caused by the first observer might affect the observations of the next observers. Furthermore, it is likely that in case an individual fish was recorded during a certain observation, it might still be there during the subsequent observation because it is foraging or hiding for predators for a specific period of time. If this period of residence is longer than the time lapse between observations, the chance that a specific observer encounters the species after it was observed during the previous observation becomes inflated (this inflated chance is the result of the non-randomness of instantaneous fish displacement). The remaining sampling variability is contained within the residual variability.

To assess the independence of successive observations and to evaluate the importance of observer effects, auto-correlation functions and binomial models were developed (section 2.2.1). To quantify sampling variability and to evaluate the methodological parameters, models allowing to assess univariate (section 2.2.3) and multivariate variability (section 2.2.4) were

developed. Transect length and number of repeats were evaluated through assessment of the design error (section 2.2.2) and precision estimates (section 2.2.5). Since independence and design error (section 2.2.1 and 2.2.2) need to be considered in model development, they were evaluated prior to the assessments of univariate and multivariate variability and precision (section 2.2.3, 2.2.4 and 2.2.5).

All analyses were performed using the R software (version 3.6.2, R Developer Core Team, R Foundation for Statistical Computing, Vienna, Austria) and the Primer v6 multivariate statistics package [50] with PERMANOVA add-on [51].

**2.2.1 Independence of observations and observer effects.** To assess whether successive observations within the same transect were independent, the auto-correlation functions (ACF; $r_{k, j}$) of the counts of each species of the 18 successive observations ($Y_{1,j}, Y_{2,j}, \ldots, Y_{i, j}, \ldots, Y_{18,j}$) per transect $j$ for lag $k$ (number of observations between two observations), were determined,

$$\bar{r_{k,j}} = \frac{\sum_{i=1}^{18-k}(Y_{i,j} - \bar{Y}_j)(Y_{i+k,j} - \bar{Y}_j)}{\sum_{i=1}^{18}(Y_{i,j} - \bar{Y}_j)^2} \tag{1}$$

In addition, for each observation lag $k$, the average ACF of the count of each species over the transects was determined, providing detailed information on the temporal scale at which repeated observations exhibited interdependence. Finally, to assess whether individual species were affected in their behavior by the presence of an observer, binomial mixed models were constructed (Linearity between predictors and the logit of the outcomes was evaluated using scatter plots). Models were constructed for the 18 successive observations per transect and for the 6 successive observations per observer and per transect. The binomial mixed models had the Presence/Absence of the concerning species ($Y'_{1,j}, Y'_{2,j}, \ldots, Y'_{18,j}$) as response, Order or sequence number of the observation $i$ as a fixed effect and transect $j$ as a random effect.

$$Y'_{i,j} \sim Binomial(n_{i,j}, p_{i,j})$$

$$\log\left(\frac{p_{i,j}}{1 - p_{i,j}}\right) = \alpha + \beta \times Order_{i,j} + b_j \times Transect_j \tag{2}$$

Species which were only observed during the first of 18 observations were likely scared off by observers, while species which were only observed during the last observations were likely attracted by the observers. Patterns for single observers could indicate that observers specifically targeted or avoided areas where species were found earlier. For these analyses, all series of observations which had at least one observed presence of the species concerned were considered. A literature-based score of reaction-to-observer was compared with the results of these models [52]. The level of independence of the observed structure of fish assemblages was assessed using a partial Mantel correlogram [53]. The correlation between (1) the number of repeats/lags between observations and (2) the Bray-Curtis dissimilarity values was determined while partialling out the effect the sampling unit (i.e. transect) and observer might have had (categorical variable included as matrix of dummy variables).

**2.2.2 Design error.** Observed absences (zeros) often have multiple origins: True zeros may either be structural, i.e. related to habitat unsuitability, environmental conditions and/or dispersal limitations; or they can be "random", i.e. related to sampling variability. False zeros may be related to counting errors, observer effects or errors in the experimental design [54]. An important design error may arise from the choice of transect length. If transect lengths are too short, many observations may contain no species at all and there may also be pairs of observations having no or only few species in common, leading to uninformative dissimilarity

values and questionable multivariate analyses [12, 50]. In those cases, lack of information within individual observations may require pooling sets of observations [55]. Therefore, to assess whether the transects were long enough to ensure that observations could be treated as solid repeats rather than subsamples, Monte Carlo simulations were performed during which percentage difference (Bray-Curtis) dissimilarity matrices were calculated [55]. Permutation scenarios ($10^4$ permutations) included the random pooling of 1 to 6 or 1 to 18 observations (mean) for different transect lengths and over two different spatial factors (i.e. Transects and Locations, respectively). The permutations and the calculation of the dissimilarity values were obtained within each level of the considered factor, allowing to assess the dissimilarity in recorded fish assemblages among different subsets of pooled observations for a given Transect or Location. Since there were 30 transects and 10 locations, 30 and 10 series of $10^4$ permutations each were conducted. Finally, the average frequency distributions of the dissimilarity values were calculated, to assess the overall trend in dissimilarity with increasing transect length and number of pooled observations.

**2.2.3 Variability in abundance and diversity estimates.** The importance of the different grouping factors (Location, Transect and Observer) for the estimates of species density and Shannon diversity were assessed using linear mixed models. To relate the counts (MaxCount) of individual species to the different factors, zero-inflated generalized linear mixed models (GLMM) were developed. The output of models with Conway-Maxwell-Poisson (COM-Poisson) distribution with log-link were reported. The COM-Poisson distribution generalizes the Poisson distribution by adding a parameter to model both underdispersion and overdispersion [56], while the negative binomial only accounts for overdispersion [57]. Since one-third of the species distributions were underdispersed and two-thirds overdispersed, the COM-Poisson distribution seemed most suitable. In addition, the count COM-Poisson models outperformed, based on AIC, count models using a Poisson or negative binomial distribution for most species. A subset of four species, with the potential to serve as indicator species for differences in the structure of fish assemblages, were selected to provide a more in-depth assessment (for a more detailed description, see S4 File). Both types of models had Island as fixed effect, Location and Transect as nested random effects and Observer as a crossed random effect. The Intraclass Correlation Coefficient (ICC) was estimated to quantify the proportion of variance explained by the different random factors [58],

$$ICC_r = \frac{\sigma_r^2}{\sum \sigma_r^2 + \sigma_\varepsilon^2}$$

$$ICC_{Total} = \frac{\sum \sigma_r^2}{\sum \sigma_r^2 + \sigma_\varepsilon^2}$$

$$ICC_\varepsilon = \frac{\sigma_\varepsilon^2}{\sum \sigma_r^2 + \sigma_\varepsilon^2} = 1 - ICC_{Total}$$

$$ICC_{Sampling} = ICC_\varepsilon + ICC_{Observer:Transect} + ICC_{Observer:Location}$$

(3)

with $\sigma_r^2$ the variance of random factor $r$ and $\sigma_\varepsilon^2$ the residual variance. $ICC_\varepsilon$ measures the proportion of unexplained variability, while $ICC_{Sampling}$ measures the proportion of sampling variability. The lmer and glmmTMB R packages were used to develop the linear and generalized linear mixed models, respectively [59].

**2.2.4 Variability in the structure of fish assemblages.** The observed fish assemblages (MaxCount) were assessed using PERMANOVA models ($10^4$ permutations under a reduced model). Data was fourth-root transformed and Bray-Curtis dissimilarities were calculated. The

models had Island as fixed factor, Location and Transect as nested random factors and Observer as crossed random factor. The addition of the factor Video Analyst was evaluated separately because of its unbalanced nature. The square root components of variation ($\sigma'$) were estimated by equating the mean squares of the PERMANOVA models to their expectations [51]. To determine the proportion of the variability explained by the different grouping factors, the percentage of variation of each component to the total variation was used (correction for different degrees of freedom among grouping factors). The combined contribution of $\sigma'_{Residuals}$, $\sigma'_{Observer:Location}$ and $\sigma'_{Observer:Transect}$ was used as a proxy of the proportion of sampling variability (see section 2.2.3). $R^2$ values (ratio of sum of squares of the grouping factors over the total sum of squares: no correction for different degrees of freedom among grouping factors) were estimated to assess the effect of longer transect lengths, different metrics (MinCount and MaxCount), data transformations (fourth root, logarithm and presence/absence) and dissimilarity indices (Bray-Curtis, Euclidean, Gower and Kulczynski) on the goodness-of-fit of the PERMANOVA models. Although the choice for a data transformation or dissimilarity index is highly dependent on the research question, researchers might still be interested in their effect on model fit. In addition, the considered transformation of abundance data to presence-absence data entails a broader discussion regarding the way data should be collected (i.e. identification and counting versus only identification respectively). Finally, following the predominant terminology used in literature, the multivariate analyses of abundance and presence/absence data will yield insights in the structure and species composition of fish assemblages, respectively. To evaluate the performance of a specific methodological parameter, all possible combinations of parameter values were tested, yielding 120 PERMANOVA models. CAP (Canonical Analysis of Principal components) was applied over the different transect lengths to assess the distinctiveness of the different Locations and Transects using the classification error after leave-one-out cross-validation [60].

**2.2.5 Precision estimates of abundance, diversity and fish assemblage structure.** Precision for both the univariate, i.e. abundance and diversity, and multivariate, i.e. fish assemblage structure, variables was estimated to determine the required number of repeats and required transect length to account for sampling variability. Univariate precision was determined as the inverse of the standard error (of the mean) to mean ratio [61]. The precision for species density, Shannon diversity and count (MaxCount) of the observed species was determined using permutations ($n = 10^4$) over the transects using the 18 observations per transect [62]. To compare the effect of more repeats and longer transect lengths, the precision was assessed as a function of the total swim distance, determined as the number of repeats multiplied by the transect length. The effect of turbidity on the precision of the species density and Shannon diversity was assessed after classifying the 30 transects in three equally sized groups based on the rank-order of the level of visually graded turbidity.

For the multivariate analysis, the square root of the residual mean square of the PERMANOVA models divided by the number of repeats was used as a measure of multivariate pseudo error variance (multSE) [55]. This proxy of multivariate precision was evaluated against transect length and number of repeats. For each transect length, permutations ($n = 10^4$) over the transects using the 18 observations per transect were applied to assess the effect of different numbers of repeats.

# 3 Results

## 3.1 Independence of observations and observer effects

Although some evidence of temporal auto-correlation was found, this was only significant for two species ($p < 0.05$), the Harlequin wrasse (*Bodianus eclancheri*) and the Pacific dog snapper

(*Lutjanus novemfasciatus*) (S1 Table). The order of the observations, determined using all 18 observations of each transect (S2 Table; mixed binomial models), had a significant negative effect on the probability of observing the Cortez rainbow wrasse (*Thalassoma lucasanum*, $p < 0.001$), Marbled goby (*Gobio manchada*, $p = 0.015$) and Stone scorpion fish (*Scorpaena mystes*, $p = 0.006$), suggesting that these fish species were scared away by observers. By contrast, the order of the observations had a significant positive effect on the probability of observing the Amarillo snapper (*Lutjanus argentiventris*, $p = 0.048$), Galapagos triplefin blenny (*Lepidonectes corallicola*, $p = 0.001$), Mexican hogfish (*Bodianus diplotaenia*, $p = 0.007$), Panamic sergeant major (*Abudefduf troschelii*, $p < 0.001$) and White salema (*Xenichthys agassizii*, $p = 0.034$), suggesting that these fish species were attracted by the observers. For the mixed binomial models using all 6 observations of each transect per observer, the order of the observations only had a significant positive effect on the probability of observing the Panamic sergeant major (*Abudefduf troschelii*, $p = 0.007$) and Reef cornet fish (*Fistularia commersonii*, $p = 0.048$) and a marginally positive effect on the Bravo clinid (*Gobioclinus dendriticus*) ($p = 0.078$). The order of the observations had a significant negative effect on the probability of observing the Blue and gold snapper (*Lutjanus viridis*, $p = 0.024$) and Yellowtail damselfish (*Chrysiptera parasema*, $p = 0.001$). For 9 out of 11 species with significant observer effects, the literature-based score of the reaction-to-observer corresponded with the observed fish behavior towards observers (Table G.2 in S1 Appendix). There was a discrepancy in the observed effects and literature-based score for the Amarillo snapper and Yellowtail damselfish. The partial Mantel correlogram indicated that temporal dependencies in the structure of fish communities were not significant ($p < 0.001$), suggesting that observations could be treated as independent.

## 3.2 Design error

Pooling observations had a clear effect on the frequency distributions of the dissimilarity values for both spatial factors, Transect and Location, and for all different transect lengths (10 to 50 meters) (Fig 3). As expected, the observations became more similar (lower dissimilarity values) when pooling observations, a trend which was most pronounced for the longer transect lengths (Fig 3). However, even at transects of 10 meters, hardly any uninformative dissimilarity values were obtained (S6 and S7 Tables). Therefore, there was no indication that the observations had to be pooled and the original observations could be used for the multivariate models. The higher number of lower dissimilarity values when pooling within Transect compared to pooling within Location was expected, since the repeated observations of a single sampling unit (Transect) are likely to be more similar than the observations of two different sampling units within the same location. Because of the clear difference between transects of the same location, it was decided to only determine the precision within Transect (over the different observers) and not within Location. Hence, per response variable, 30 (=number of transects) precision curves could be constructed (see section 3.5).

## 3.3 Variability in abundance and diversity estimates

The $ICC_{Total}$, representing the proportion of the variability of the species density and Shannon diversity explained by the factors Location, Transect, Observer and their interaction increased with increasing transect length (Table 2). For both the species density and Shannon diversity the proportion of sampling variability decreased with increasing transect length. For species density, there was a slight decrease in the proportion of variability explained by Location, but this was compensated by the increase in the proportion of variability explained by Transect. The factors Observer and interaction Observer:Location explained none or only little of the

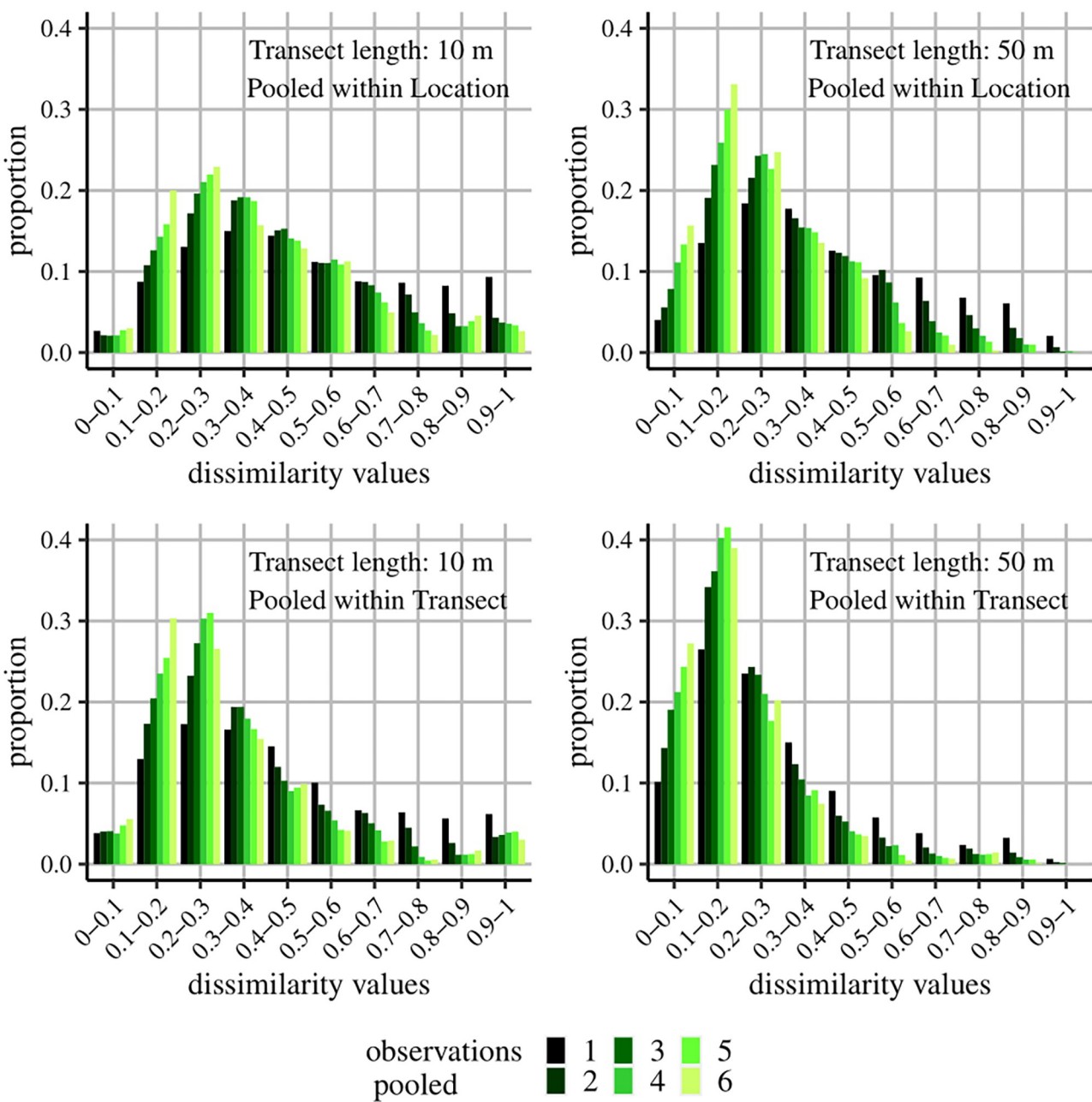

**Fig 3. Pooling observations.** Histograms representing the frequency distributions of dissimilarity values (fourth-root transformed, Bray-Curtis) for different pooling scenarios. Low dissimilarity values indicate that observations are relatively similar. Different spatial scales (Location versus Transect), Transect lengths (10 versus 50 meters) and number of observations to be pooled (1 to 6) were assessed. As observations within transects and locations are compared, we would expect relatively low dissimilarity values.

variability of the species density, but the proportion of variability explained by the interaction Observer:Transect was relatively high. For a transect length of 10 and 20 meters it was even higher than the proportion of variability explained by Transect.

Similarly, the proportion of variability of the Shannon diversity explained by Transect increased with increasing transect length (Table 3). The proportion of variability explained by Location decreased rapidly, while the proportion of variability explained by the interactions

**Table 2. Model of the observed species density.** Model output (ICC (Intraclass Correlation Coefficient) of the random effects, total ICC explained by the factors of the model and ICC associated with the sampling variability) for a linear mixed model with species density as response, Island as fixed effect, Location and Transect as nested random effects and Observer as crossed random effect. Transect lengths of 10, 20, 30, 40 and 50 meters were assessed.

| | 10 meters | 20 meters | 30 meters | 40 meters | 50 meters |
|---|---|---|---|---|---|
| $ICC_{Transect}$ | 0.073 | 0.116 | 0.145 | 0.180 | 0.204 |
| $ICC_{Location}$ | 0.369 | 0.354 | 0.345 | 0.351 | 0.345 |
| $ICC_{Observer}$ | 0.000 | 0.003 | 0.010 | 0.010 | 0.007 |
| $ICC_{Observer:Transect}$ | 0.103 | 0.131 | 0.092 | 0.112 | 0.099 |
| $ICC_{Observer:Location}$ | 0.015 | 0.000 | 0.004 | 0.000 | 0.007 |
| $ICC_{Total}$ | 0.560 | 0.604 | 0.596 | 0.653 | 0.662 |
| $ICC_{Sampling}$ | 0.558 | 0.527 | 0.500 | 0.459 | 0.444 |

Observer:Transect and Observer:Location showed a clear increase. The latter were even higher than the proportion of variability explained by Location from transect lengths 20 to 50 meters.

The proportion of variability of the counts of the potential indicator species explained by the factors Location, Transect, Observer and their interaction did not show any consistent trend with increasing transect length (S4 Table). The proportion of variability explained by Observer (i.e. observer bias) was lower than the proportion of variability explained by Transect and Location for 34 out of 36 species. However, interaction effects of Observer with either Transect or Location were often important as the proportion of variability explained by the interactions Observer:Transect and Observer:Location, exceeded the proportion of variability explained by Transect and Location for 18 out of 36 species. For 28 out of 36 species the sampling variability exceeded the variability explained by Transect and Location (S5 Table).

## 3.4 Variability in the structure of fish assemblages

For all different transect lengths, the factors Island, Location and Transect explained a larger amount of the observed variability than the factor Observer and its interactions (largest proportion of $\sigma'^2$: estimates of components of variation) in the PERMANOVA models ($10^4$ permutations) (Table 4). The factor Observer only explained a limited proportion of the variability (between 0 and 0.3%). However, a relatively large part of the variability (between 0.9 and 4.4%) was explained by the interactions of Observer with Location and Observer with Transect. With increasing transect length the proportion of sampling variability decreased (from 25.9 to 17.4%), while the proportion of variability explained by the factors Location and Transect increased (from 8.2 to 12.5% and from 10.0 to 12.8% respectively). Inclusion of the factor Video Analyst had a non-significant and negligible effect (S9 Table).

**Table 3. Model of the observed Shannon diversity.** Model output (ICC (Intraclass Correlation Coefficient) of the random effects, total ICC explained by the factors of the model and ICC associated with the sampling variability) for a linear mixed model with Shannon diversity as response, Island as fixed effect, Location and Transect as nested random effects and Observer as crossed random effect. Transect lengths of 10, 20, 30, 40 and 50 meters were assessed.

| | 10 meters | 20 meters | 30 meters | 40 meters | 50 meters |
|---|---|---|---|---|---|
| $ICC_{Transect}$ | 0.184 | 0.261 | 0.380 | 0.420 | 0.442 |
| $ICC_{Location}$ | 0.104 | 0.044 | 0.000 | 0.000 | 0.000 |
| $ICC_{Observer}$ | 0.001 | 0.000 | 0.009 | 0.008 | 0.010 |
| $ICC_{Observer:Transect}$ | 0.082 | 0.119 | 0.066 | 0.078 | 0.161 |
| $ICC_{Observer:Location}$ | 0.065 | 0.081 | 0.081 | 0.110 | 0.105 |
| $ICC_{Total}$ | 0.436 | 0.505 | 0.536 | 0.616 | 0.718 |
| $ICC_{Sampling}$ | 0.711 | 0.695 | 0.611 | 0.572 | 0.548 |

**Table 4. Multivariate variation of the fish assemblage structure.** Square root estimates of components of variation ($\sigma'$) and the percentage of variation of each component to the total variation of PERMANOVA models based on Bray-Curtis dissimilarities (fourth-root transformed) with Island as fixed factor, Location and Transect as nested random factors and Observer as crossed random factor for transect lengths of 10, 20, 30, 40 and 50 meters.

| | 10 meters | 20 meters | 30 meters | 40 meters | 50 meters |
|---|---|---|---|---|---|
| $\sigma'_{Island}$ | **45.70 | **43.13 | **41.66 | **40.21 | **39.39 |
| $\sigma'_{Observer}$ | 3.17 | -0.42 | 1.99 | 2.31 | 2.72 |
| $\sigma'_{Location}$ | **17.58 | **17.49 | **17.47 | **18.02 | **18.41 |
| $\sigma'_{Island:Observer}$ | 1.40 | 2.82 | 2.37 | 2.46 | 1.25 |
| $\sigma'_{Transect}$ | **19.35 | **19.60 | **19.54 | **19.27 | **18.61 |
| $\sigma'_{Location:Observer}$ | *5.84 | **8.34 | **8.14 | **8.18 | **8.98 |
| $\sigma'_{Transect:Observer}$ | **11.90 | **12.34 | **11.53 | **11.15 | **10.92 |
| $\sigma'_{Residuals}$ | 28.19 | 23.28 | 20.66 | 18.58 | 16.49 |
| % $\sigma'^2_{Island}$ | 55.64 | 56.00 | 56.76 | 56.52 | 57.09 |
| % $\sigma'^2_{Observer}$ | 0.27 | 0.00 | 0.13 | 0.19 | 0.27 |
| % $\sigma'^2_{Location}$ | 8.23 | 9.21 | 9.98 | 11.35 | 12.48 |
| % $\sigma'^2_{Island:Observer}$ | 0.05 | 0.24 | 0.18 | 0.21 | 0.06 |
| % $\sigma'^2_{Transect}$ | 9.97 | 11.57 | 12.48 | 12.98 | 12.75 |
| % $\sigma'^2_{Location:Observer}$ | 0.91 | 2.09 | 2.17 | 2.34 | 2.97 |
| % $\sigma'^2_{Transect:Observer}$ | 3.77 | 4.59 | 4.34 | 4.34 | 4.40 |
| % $\sigma'^2_{Residuals}$ | 21.16 | 16.31 | 13.95 | 12.07 | 10.01 |
| % $\sigma'^2_{Sampling}$ | 25.89 | 23.23 | 20.64 | 18.96 | 17.44 |

*$p < 0.05$,

**$p < 0.01$

Using different transect lengths, metrics, data transformations and methods to calculate dissimilarity matrices had a significant ($p < 0.05$) effect on the goodness-of-fit of the PERMANOVA models (pairwise paired t-test of $R^2$ values of each level of every considered factor with Bonferroni correction. Interactions were not considered). The most significant differences were due to the increase of the transect length followed by the method to calculate the dissimilarity matrices (Bray-Curtis, Euclidean, Gower and Kulczynski), the metrics (MaxCount and MinCount) and finally the data transformation (fourth-root, logarithm and presence/absence) (S7 and S8 Figs).

The leave-one-out cross-validation classification error of the CAP analysis decreased with increasing transect length for both the grouping factors Transect and Location (Table 5). The decrease was most pronounced for the analysis using Transect as grouping factor.

Given the pronounced difference of the fish assemblage structure between islands, the first PCO axis of a PCO analysis of the full data was a good proxy to distinguish the fish assemblage structures of both islands. Using the correlations of the full data with this first PCO axis, the

**Table 5. Classification error of leave-one-out cross-validation of CAP analyses (Bray-Curtis, fourth-root transformed, eight PCO axes ($m$)).** CAP analyses were performed for different transect lengths (10, 20, 30, 40 and 50 meters) and different grouping factors (Location and Transect).

| | Leave-one-out cross-validation classification error | | | | |
|---|---|---|---|---|---|
| | 10 meters | 20 meters | 30 meters | 40 meters | 50 meters |
| Location | 35.37 | 36.11 | 28.70 | 25.56 | 25.00 |
| Transect | 60.37 | 52.78 | 46.85 | 42.04 | 36.85 |

species that are most important to explain this difference between islands, could be identified. The most important species to explain differences in fish assemblage structure between the islands included, for Santa Cruz, the Bullseye puffer fish (*Sphoeroides annulatus*; $r = -0.62$), Yellowtail damselfish (*Microspathodon bairdii*; $r = -0.83$) and Pacific spotfin mojarra (*Eucinostomus dowii*; $r = -0.64$), and, for Floreana, the Bravo clinid (*Gobioclinus dendriticus*; $r = 0.56$), Galapagos ringtail damselfish (*Stegastes beebei*; $r = 0.83$) and Chameleon wrasse (*Halichoeres dispilus*; $r = 0.87$).

## 3.5 Precision estimates of abundance, diversity and fish assemblage structure

Precision, defined as the inverse of the standard error over the mean, of our estimates of both species density and Shannon diversity increased with increasing number of observations and increasing transect length (Fig 4A and 4C). Although the precision improved markedly with increasing total swim distance (transect length x number of repeats), it did not matter much whether more repeats or longer transects were considered: 10-meter transects were slightly more efficient to obtain precise results on the species density (Fig 4B), while 20 or 30-meter transects were slightly more efficient for the Shannon diversity (Fig 4D). For both diversity measures, the precision seemed to level off at approximately 250 meters of total swim distance.

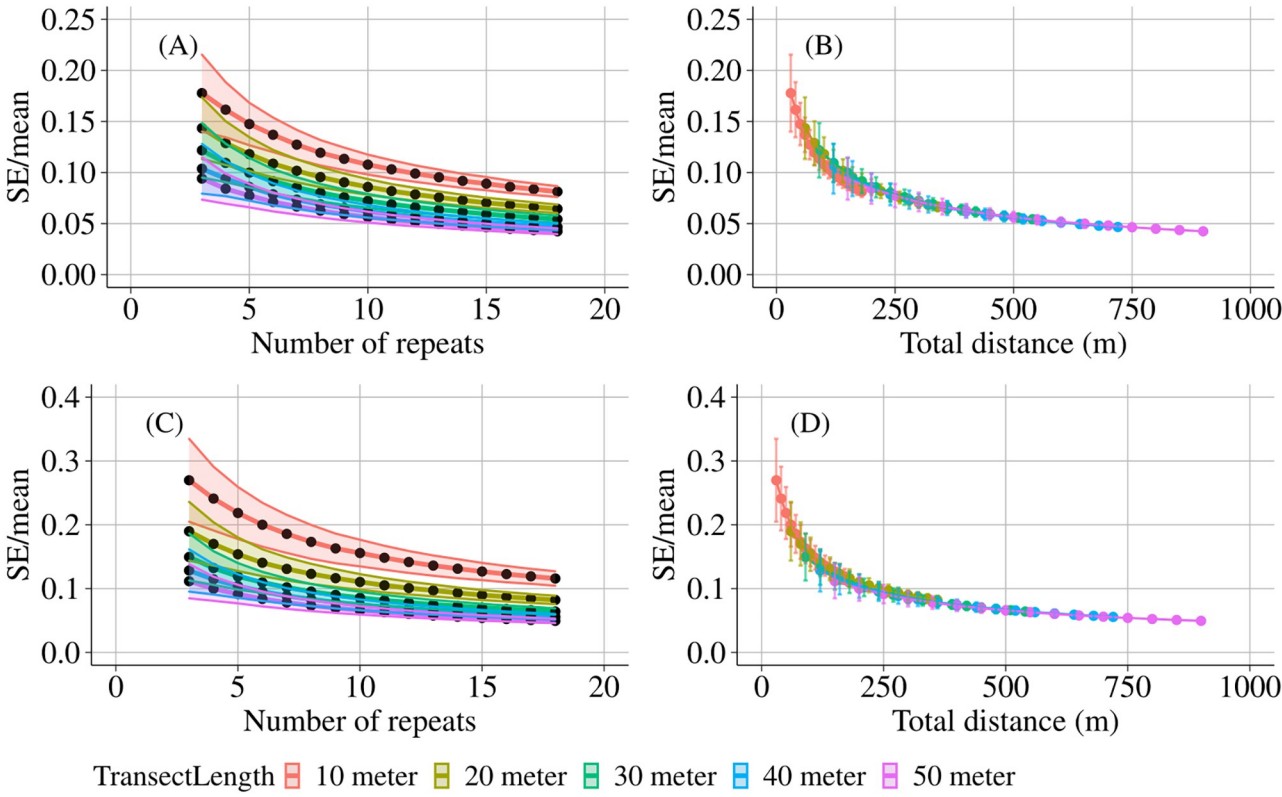

**Fig 4. Univariate precision estimates of species density and Shannon diversity.** Estimates of the standard error over the mean for species density (A) and Shannon diversity (C) in function of the number of repeats. Precision is defined as the inverse of the standard error over the mean. In (B) and (D) the standard error over the mean of species density and Shannon diversity is given as a function of the total swim distance. Total swim distance is defined as the number of repeats multiplied by the transect length. Different transect lengths were considered ranging from 10, 20, 30, 40 to 50 meters. Monte Carlo simulations ($n = 10^4$) were applied to determine the standard error over the mean per transect. The average values over all transects are visualized. The error bars represent the 95% confidence intervals which were constructed using the pooled standard deviation of the estimates.

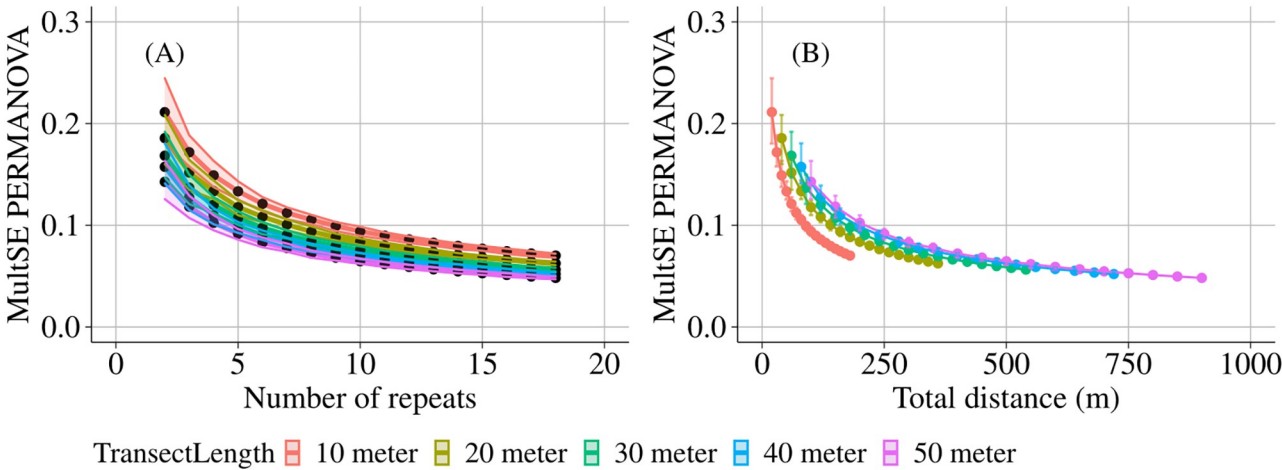

**Fig 5. Multivariate precision estimates of fish assemblage structure.** Multivariate pseudo error variance (multSE) of PERMANOVA analysis based on Bray-Curtis dissimilarities (fourth-root transformed) with Island as fixed factor and Location and Transect as nested random factors. Monte Carlo simulations (n = $10^4$) were applied to determine the multSE per number of repeats. The average values over all transects are visualized. The error bars represent the 95% confidence intervals which were constructed using the pooled standard deviation of the estimates.

Higher levels of turbidity caused lower levels of precision for both the species density and Shannon diversity (S9 Fig). For all observed species, shorter transects (10 and 20 meters) with more repeats turned out to be most efficient to improve precision in terms of total swim distance, with the exception of the Galapagos Sea Bream (*Archosargus pourtalesi*) and the Black Striped Salema (*Xenocys jessiae*), for which longer transect lengths and less repeats were more efficient (S10 Fig). The stronger effect of repeats over transect lengths was more clear for the counts of the individual species than for the diversity indices, but the total swim distance at which the precision curve levelled off was highly species dependent. Multivariate pseudo error variance (MultSE) was also clearly affected more strongly by the number of repeats than by transect length, hence choosing 10-meter transects turned out to be most efficient to improve precision (Fig 5). MultSE leveled off at longer total swim distances for the longer transect lengths; for example, for 50-meter transects this was around 350 meters, while for 20-meter transects this was around 200 meters.

## 4 Discussion

### 4.1 Observer bias, observer effects and instantaneous fish displacement

Observer biases and sampling variability are important aspects to consider in observational studies of fish assemblages, but the simple sampling designs and lacking replicates of most studies hampers a detailed assessment [36, 63]. The hierarchical sampling design of this study allowed to estimate inter-observer variability related to the observers themselves (i.e. observer bias) and their interaction with the sampling units and sites (i.e. observer effects and instantaneous fish displacement). Although the provided estimates of the observer bias and sampling variability can be considered as relatively reliable, the additional partitioning of sampling variability in observer effects, instantaneous fish displacement and random counting/detection errors is subjected to much more uncertainty. Although fish displacement at very short time scales is often referred to as random, perfect randomness seldomly occurs in ecology and undetected spatiotemporal patterns are more likely the rule than the exception (e.g. undocumented foraging and reproduction behavior and the conditional nature of animal movement

in terms of space and time) [64]. Since patterns related to observer effects and instantaneous fish displacement can therefore be confounded and difficult to unravel, additional insights in the biological and ecological traits of fish can be useful to distinguish between causes of variation [13, 47, 65].

The relatively low $ICC_{Observer}$ values, which are an indication of the proportion of variability explained by the factor Observer, of most univariate models in combination with the low $\sigma'_{Observer}$ of the multivariate models suggested that systematic observer biases were generally limited. However, the interaction between the factor Observer and the factor Transect or Location often turned out to be important, suggesting that some more complex processes involving observer effects and instantaneous fish displacement might be involved. This corroborates with the results of visual census studies in which observer biases were less important than sampling variability [18, 45]. For the observed structure of the fish assemblages and the diversity estimates, i.e. species density and Shannon diversity, these interaction effects were considerably smaller than the effects attributed to the factors Location and Transect. However, for the univariate species count models, 18 out of 36 species had considerably high inter-observer variabilities compared to the effects attributed to the factors Location and Transect. Observers covered the transects sequentially, meaning that each observer first finished six observations of one transect before moving to another transect. Therefore, one would expect that observations closer in time would be more similar, causing significant interaction effects of the factors Transect and Observer. Although mostly not significant, there were indeed some clear patterns in temporal auto-correlation for some species, with similar amounts of species being significantly deterred (5) or attracted (7) by observers. In addition, differences in detectability due to intrinsic factors of fish species such as crypsis and color or extrinsic factors such as observer behavior may have caused an additional bias [15, 47, 66]. For example, the Marbled goby *(Gobio manchada)*, a shy and cryptic species that hides in crevices, was scared away by observers, while the Panamic sergeant major *(Abudefduf troschelii)* and Mexican hogfish (*Bodianus diplotaenia*), two species which are known to not be frightened easily, appeared attracted to observers as they were typically recorded more often during the first and last observations, respectively [52]. It was unlikely that individuals of scared-away fish species would be recorded by the second and third observer, while non-scared fish would be more likely to be observed by all observers. Indeed, models for fish that were scared away typically showed higher levels of inter-observer variability than models for fish that seemed to be attracted to the observer.

Besides the effect of observers on fish behavior, there are also effects of fish behavior on observer perception. Schooling behaviour may cause higher inter-observer variability [16, 19]. As a single observer records a school of fish that moves out of view during the subsequent observations of other observers surveying that specific transect or location, the effect on inter-observer variability would be much larger than when a single individual would move out of view. This was most likely the case for the White salema (*Xenichthys agassizii*) and Galapagos grunt (*Orthopristis forbesi*) which are known to spatially cluster and move around in group [52]. Similarly, the displacement of individual fish in and out of view will have a stronger effect on the inter-observer variability of the counts of rare species (such as the Jewel moray (*Muraena lentiginosa*)) compared to that of more common species, as instantaneous fish displacement will affect observed fish abundances more severely [52], potentially even causing species to be detected during only some of the observations of a specific transect or location [66].

Finally, the behavior of an observer might also affect his/her perception. Individual observers covering a transect multiple times may gain knowledge regarding the location of species that were hiding and may unconsciously give these locations more attention. For example, on the level of a single observer, once the sedentary Bravo clinid (*Gobioclinus dendriticus*) was

detected [52], it was much more likely that it would also be detected during successive observations. Therefore, the interaction effects are likely the result of a combination of temporal autocorrelation between observations, instantaneous fish displacement and biases related to traits of both fish and observer.

Overall, for the multivariate structure of the fish assemblages, the factor Observer and its interaction with spatial factors turned out to be much less important than the factors Location and Transect. This result is in strong contrast with the results of traditional visual census studies which are often faced with high levels of inter-observer variability [9, 29, 36, 45]. Although, the "unwanted" sampling variability associated with instantaneous fish displacement, observer effects and random counting/detection errors could be relatively large for short transects (1.42 times the variability explained by transects and locations), longer transects allowed for a significant improvement (0.69 times the variability explained by transects and locations). Although not assessed in this study, detectabilty of species has been shown to be positively related to density [67]. Hence, the choice of the transformation and/or dissimilarity measure, which differ in the weights given to rare and common species, might also affect the importance of observer effects and instantaneous fish displacement. In addition, if researchers are interested in the distributions of individual species, species-specific observer effects and instantaneous fish displacement might still be very important as the proportion of sampling variability of most species (i.e. 28 out of 36 species for transect lengths of 50 meters) exceeded the proportion of variability explained by the sampling units and sites.

The overall low observer bias suggests that one observer is sufficient and that observations of different observers under similar conditions are interchangeable, given these observers received a similar training. Training can limit counting/detection errors and observer effects to some extent, but an important part of the sampling variability related to the observer effects, counting/detection errors, and instantaneous fish displacement will remain. Although this "uncontrollable" yet important part of the sampling variability cannot be reduced, it can be accounted for when sufficient repetitions of sufficiently long transects are available (see section 4.2) [8, 20].

## 4.2 Design error and sampling variability

The definition of a design error depends on the hypothesis and on the scale of a study. For example, if one is interested in the association of a species or fish assemblage with different classified habitat patches, the size of the repeated sampling unit is likely to affect the sampling variability [12]. If instead, one is interested in the association of a species or fish assemblage with a specific area, sampling units that are too small may introduce a design error because of their inability to account for the range of different habitat types present in the study area [68].

Although habitat patches were not explicitly classified in this study, each transect could be thought of as a unique series of habitat patches that did not alter during sampling. To assess whether species or fish assemblages are effectively associated with a specific transect (or subset of a transect), some repeated observations are required. Precision quantifies the concordance of multiple observations and was determined for species density, Shannon diversity and counts of individual species of each transect. The precision of most parameters increased with the number of repeats and the transect length. Hence, longer transect lengths were characterized by lower sampling variabilities. In addition, the lower CAP cross-validation classification errors (Table 5) and the lower dissimilarity values of the longer transects (Fig 3) suggested that also for a multivariate response (e.g. fish assemblage structure), sampling variability decreased with increasing sampling length.

In addition to associations with habitats, researchers are often interested in associations of fish species or assemblages with specific areas or locations. Instead of the assessment of small-scale relationships between fish and their environment, one then wants to obtain a more general idea of how fish are affected by their surrounding environment [69]. Such an approach makes sense, given the high mobility of different fish species and the often strong spatiotemporal dynamics of environmental conditions in coastal areas [70]. Therefore, rather than assessing the sampling variability within transects, the variability between transects and between locations should be assessed. The linear mixed models of species density and Shannon diversity fitted the data better with increasing transect length, which is likely a consequence of the smaller chance of discovering additional species with further sampling effort in a specific location or transect, as can be seen in the species accumulation curves of S4–S6 Figs. Similarly, the PERMANOVA models fitted the data better (Table 4) and CAP cross-validations had lower classification errors for the different locations (Table 5) with increasing transect length. If more different micro-habitats are monitored, the observed fish assemblages within a loca-tion will likely be more representative for that location (S8 Fig). The improved goodness-of-fit of the PERMANOVA models with increasing transect length ($R^2_{10m} = 0.761$; $R^2_{50m} = 0.880$), could mainly be attributed to the factors Location ($R^2_{Location;10m} = 0.145$; $R^2_{Location;50m} = 0.212$) and to a lesser extent to Transect ($R^2_{Transect;10m} = 0.117$; $R^2_{Transect;50m} = 0.142$), suggesting that, in addition to the reduced sampling variability within transects, the representativeness of the transects for a specific location improved. The Observer-related factors of the PERMANOVA models remained limited, suggesting that, unlike the sampling variability and design error, the inter-observer variability did not change much from short to long transects (Table 4). In addition, the design error associated with short transects did not appear to impede multivariate analyses of the assemblage structure for the transect lengths considered in this study: Even the lowest transect length of 10 meters provided interpretable data for analyses of the assemblage structure as uninformative dissimilarity values were almost non-existent, indicating that there was no need to pool samples (Fig 3).

Not only the procedures to collect video material, but also the procedures to analyze them should be evaluated. The PERMANOVA model using log-transformed MaxCount data from 50-meter transects with six repeats performed best, while the PERMANOVA models using presence-absence or transformed MinCount data of lower transect lengths performed worst (S7 Fig). MaxCount models were significantly better in explaining the variability than the Min-Count models, probably because the latter may not scale linearly with true abundance [71]. MinCount is often used instead of MaxCount in video monitoring studies to avoid double counts [71]. However, since we used video transects rather than stationary devices, the risk of double counting was considerably less. Although neglecting the abundance of a species scored worst in terms of goodness-of-fit, the difference between MinCount and presence-absence data was rather small. Therefore, in this specific case, if one wanted to reduce the time required for video analysis, the presence-absence instead of the abundance data could have been used, without much information being lost.

Transect length remained the most important parameter affecting the performance of the PERMANOVA models, but longer transects also increased the time required for both the data collection and processing. The square root of the residual mean square of the PERMANOVA analysis decreased with increasing number of repeats and with increasing transect length. This proxy for multivariate precision was more affected by number of repeats than by transect length, suggesting that more repeats are preferred over longer transects to increase the precision. However, more repeats do not alter the representativeness of the sampling unit for the study area. Therefore, the choice between more repeats or longer transects should be guided

by the required level of precision and representativeness of the sampling unit for the study area.

The precision of the MaxCount of most encountered species, with the exception of the Galapagos Sea Bream (*Archosargus pourtalesi*) and the Black Striped Salema (*Xenocys jessiae*), was also affected more strongly by the number of repeats than by transect length. Although there was a clear increase in the precision of species density and Shannon diversity with total swim distance, it did not matter much whether longer transect lengths or more repeats were applied. For the assessment of species density, 10-meter transects were slightly more efficient, while for the Shannon diversity, 20 to 30-meter transects were slightly more efficient. In addition to the number of repeats and the transect length, the visibility also alters the precision [47, 72, 73]. A higher turbidity resulted in a lower visibility which caused clearly lower precision levels of the species density and Shannon diversity. Observers had to swim longer transect lengths or perform more repeats in turbid conditions compared to clear water conditions. Overall, if researchers want to improve precision because of low visibility or pronounced spatial and temporal dynamics, more repeats should be preferred over longer transects. The improved ability to account for the existing sampling variability when including more repeats is stronger (per unit of sampling effort) than the decrease in sampling variability when using longer transects.

It should be noted that, besides the objectives of the study, time constraints and the nature of the parameter of interest, the most optimal distribution of the sampling effort among different methodological aspects may also be affected by more local logistic constraints [12]. For example, in order to choose between more repeats and longer transects, researchers need to balance the expected degree of representativeness, precision and disturbance tied up with the sampling units. On the one hand, it may be more difficult to maintain homogeneous areas with respect to the observed fish assemblages when transects are longer [55], but on the other hand, more repeats of the same transect might cause more disturbances and may be less informative when the objective is to obtain representative observations of the local fish assemblages. More and longer transects would in that case be more suitable.

## 5 Conclusion

Deciding on the distribution of the sampling effort among different methodological aspects is an important step for any study. In this study we described how researchers should take into account a broad range of potentially conflicting theoretical and practical aspects when setting up an experimental design for video transects. For most species, the presence of an observer has a significant effect on fish behavior which introduces severe dependence between subsequent observations. These observer effects seem however negligible when the overall fish assemblage structure is assessed rather than the counts of individual species. Similarly, the overall sampling variability (consisting of observer effects, instantaneous fish displacement and random counting errors) can for some individual species be very high (up to 100%), while for community assessments it remains relatively low (17–26%). The errors associated with the perception of the observers themselves on the other hand, i.e. observer bias, is negligible for both community and species assessments (mostly below 1%). Longer transects have a much lower sampling variability (33% lower for 10 to 50 meter transects for community assessments), yet to increase precision of repeated observations, more repeats seem more effective than longer transects. For transects, the MaxCount is prefered over MinCount as the former seems to yield a lower sampling variability even though the video analysis time is the same. Using counts instead of presence/absence data is also prefered, as the sampling variability of the observations of the former is somewhat lower. However, if the amount of time available for

video analysis is limited, presence/absence observations still provide observations of reasonably low sampling variability. In summary, sampling variability, methodological parameters, environmental conditions, and ecological knowledge, e.g. species-specific traits affecting detectability, are important for video transects and should be accounted for to properly assess the ecological state and the effect of management practices on the environment.

## Supporting information

**S1 File. Instantaneous fish displacement.**
(PDF)

**S2 File. Applications of video monitoring.**
(PDF)

**S3 File. Guidelines for the video analysis.**
(PDF)

**S4 File. Indicator species.**
(PDF)

**S1 Fig. Map of the study area.**
(PDF)

**S2 Fig. Temporal autocorrelation of observed community structure between observations without sample effect.**
(PDF)

**S3 Fig. Temporal autocorrelation of observed community structure between observations without sample effect and observer id effect.**
(PDF)

**S4 Fig. Species Accumulation Curves (SACs) in function of number of repeats.**
(PDF)

**S5 Fig. Species Accumulation Curves (SACs) in function of number of repeats.**
(PDF)

**S6 Fig. Species Accumulation Curves (SACs) in function of distance covered.**
(PDF)

**S7 Fig. Models of the fish assemblage structure using different parameters.**
(PDF)

**S8 Fig. Model of the fish assemblage structure for the full data.**
(PDF)

**S9 Fig. Precision estimates of species density and Shannon diversity.**
(PDF)

**S10 Fig. Precision estimates of observed abundance of different potential indicator species.**
(PDF)

**S1 Table. Temporal autocorrelation of observed abundance of single species between observations.**
(PDF)

**S2 Table. Observer effects on observed presence/absence of single species.**
(PDF)

**S3 Table. Selection of potential indicator species.**
(PDF)

**S4 Table. Models of the abundance of single species, for different transect lengths.**
(PDF)

**S5 Table. Models of the abundance of single species.**
(PDF)

**S6 Table. Pooling of observations within transects.**
(PDF)

**S7 Table. Pooling of observations within locations.**
(PDF)

**S8 Table. Models of the fish assemblage structure using different transect lengths.**
(PDF)

**S9 Table. Models of the fish assemblage structure using different parameters.**
(PDF)

**S1 Appendix.**
(PDF)

## Author Contributions

**Conceptualization:** Stijn Bruneel, Rafael Bermudez, Peter Goethals.

**Data curation:** Stijn Bruneel.

**Formal analysis:** Stijn Bruneel.

**Funding acquisition:** Rafael Bermudez, Peter Goethals.

**Investigation:** Stijn Bruneel, Long Ho, Amber Schoeters, Heleen Raat, Peter Goethals.

**Methodology:** Stijn Bruneel, Rafael Bermudez, Peter Goethals.

**Project administration:** Rafael Bermudez, Peter Goethals.

**Resources:** Rafael Bermudez, Peter Goethals.

**Software:** Stijn Bruneel.

**Supervision:** Rafael Bermudez, Peter Goethals.

**Visualization:** Stijn Bruneel.

**Writing – original draft:** Stijn Bruneel.

**Writing – review & editing:** Stijn Bruneel, Long Ho, Wout Van Echelpoel, Amber Schoeters, Heleen Raat, Tom Moens, Rafael Bermudez, Stijn Luca, Peter Goethals.

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
