## [Decision Letter · Decision Letter 0]

1 Jul 2021

PONE-D-21-06060

Sampling errors and variability in video transects for assessment of reef fish assemblage structure and diversity

PLOS ONE

Dear Dr. Bruneel,

Thank you for submitting your manuscript to PLOS ONE. After careful consideration, we feel that it has merit but does not fully meet PLOS ONE’s publication criteria as it currently stands. Therefore, we invite you to submit a revised version of the manuscript that addresses the points raised during the review process.

We look forward to receiving your revised manuscript.

Kind regards,

Athanassios C. Tsikliras

Academic Editor

PLOS ONE

Journal Requirements:

2. In your Methods section, please provide additional location information of the study sites, including geographic coordinates for the data set if available.

Reviewers' comments:

Reviewer's Responses to Questions

**Comments to the Author**

1. Is the manuscript technically sound, and do the data support the conclusions?

Reviewer #1: Partly

Reviewer #2: Yes

2. Has the statistical analysis been performed appropriately and rigorously? 

Reviewer #1: Yes

Reviewer #2: Yes

3. Have the authors made all data underlying the findings in their manuscript fully available?

Reviewer #1: Yes

Reviewer #2: No

4. Is the manuscript presented in an intelligible fashion and written in standard English?

Reviewer #1: No

Reviewer #2: Yes

5. Review Comments to the Author

Reviewer #1: Overview

This paper reports on the accuracy, precision and repeatability of visual transects of shallow water reef fish recorded with underwater video. In particular, the authors aim to address whether there were species-specific observer affects, the importance of observer bias sampling variability, and the importance of transects length number repeats the counting metrics and the way that data was transformed.

As an initial comment the authors do not recognise that the level of transformation that you use is largely dependent on the ecological question that you are asking.

I also note that they are missing some of the fundamental literature including previous similar work and recommend that they actually read the reference below which is fundamental to this field of study.

McCormick, M.I., Choat, J.H. Estimating total abundance of a large temperate-reef fish using visual strip-transects. Mar. Biol. 96, 469–478 (1987). https://doi.org/10.1007/BF00397964

I found the paper extremely long and overly complicated. Also, while I recognise that the statistics used are valid, I am not convinced they are the optimal methods for conveying and supporting the authors interpretation presented in the discussion.

I’m also concerned that the authors may have unwittingly introduced another source of bias into their design which may well greatly impact their analysis and data collection.

At lines 144 to 145 the authors describe laying out a transact rope and recording 2.5 m either side of the transact. Did they delineate the whole transect by laying out a rectangle or did they lay out a rope down the centre and then the estimated the distance either side of that transect by the image analyst? If they did the latter can the author’s please clearly define how they actually estimated distance. Distance estimates are 1 of the main sources of error that come into fish transects as it can be extremely difficult to consistently estimate distance. As a consequence, an observer may greatly over or underestimate the actual area observed.

See Harvey E, Fletcher D, Shortis MR, Kendrick GA. A comparison of underwater visual distance estimates made by scuba divers and a stereo-video system: implications for underwater visual census of reef fish abundance. Marine and Freshwater Research. 2004 Sep 23;55(6):573-80.

The authors also talk about a standard operating procedure developed by the aquatic ecology research group of Ghent University. It would be really useful if this could be deposited as an online appendix so that people could refer to that. Also, how does that align with already published standard operating procedures for doing video transects surveys of fish?

I also believe there are some real fundamental errors with this paper in that the authors are trying to count mobile fishes that are quite large and also small fishes such as travel fins and Lenny’s which are habitat associated. Most fish ecologists would use transects of different sizes depending on the mobility and size of the species that they are trying to investigate. Trying to sample them all in one survey is probably not realistic or ecologically valid. I certainly would not be trying to survey fish such as the Galapagos triplefin blenny (max size 9cm and highly territorial) in the same transects that I would be targeting the Amarillo snapper (a species up to 71 cm in length and highly mobile).

It would be great if the authors can make some comment about why they have lumped the species together when most guidance on surveying reef fish would recommend keeping them separate.

Overall I think the paper has the potential to be a valid contribution to science, but it needs to be substantially rewritten so that the text and the data presented a simplified. At the moment it is overly and unnecessarily complex. I recommend simplifying the analysis and the presentation of results so that readers can comprehend it and absorb the main takeaway messages. I have included a number of minor comments below.

Abstract

Line 6. I recommend that you change should to could given that this work (admittedly not for video sampling) is a repeat of past research.

Introduction

Line 56. I recommend that you write this as In the literature.

Line 62. I recommend that the authors put a reference about underwater visual census and some species being scared away and not been detected.

Lines 59-67. I commend the authors for this distinction which is actually really important.

Line 78 to 80. This statement is actually incorrect. Because of the bait plume from stationary cameras (and I am assuming you are talking about baited remote underwater video systems given you refer to Mike Cappo’s research above) have been demonstrated to be a more effective tool sampling across broad habitats and depth scales than underwater visual census. They tend to sample more species, with greater statistical power than underwater visual census. However, the bias of baited remote underwater video systems are acknowledged and one of the things that you cannot do is get density or biomass estimates per unit area.

They are also not good for getting counts of some of the smaller and less mobile species. I think you probably want to either reword this sentence or alter it to reflect a true statement.

Similarly, I think in the introduction you need to talk about the types of information that ecologists and managers want to collect and use. You have not talked all about the use of length data which some ecologists would consider to be more fundamentally useful than count data. This aspect should probably be brought into the discussion as well.

Can you provide a reference for line 87 please.

Lines 171-174. Can the authors please justify why they have used Max count or min count? One of the great advantages of transects is that you can actually obtain a density estimate. By counting all the fish that you see with in a transact you may increase your variability, but you will also increase data on length frequency. This is a fundamental advantage of doing transects over stationary camera systems. Max count is a term that comes from the baited underwater video literature and is probably inappropriate in this context. Essentially all you are doing is counting all the species and individuals that you see in a transect? I would clarify that and change your figures accordingly.

Somewhere in your methods you need to tell the reader how you analysed your video imagery and why used that particular method. At the moment, someone wanting to repeat your work couldn’t do it as that information is not in the main text or the appendices.

Line 179 to 181. Please make a comment about what the effect of not being a to identify the exact length of the transects might have on the data. It seems like this is probably going to increase a source of error and variability as you may be greatly increasing or decreasing the area surveyed.

Line 190. Do you have an example or a reference supporting your thoughts about instantaneous fish displacement? Has anyone else observed this in the scientific literature?

Figure 3. In the methods I think it would be really useful of the authors stated why they have used the similarity values to represent their transects and not looked at the cumulative number of species or individuals and also presented the means. The dissimilarity values are a valid way of presenting this, but an unusual way of presenting this data. I’m not convinced that this the optimal way of presenting it.

Lines 429 - 442. Rather than using the components of variation have the authors considered using a distant based linear model which might be easier for the reader to understand?

Line 464. Please change the use of species density as if you are using min count or Max count you’re not counting all the individuals of a species. Again, you have used Max count inappropriately.

Line 533 - 534. Please note that both the species that you list as pelagic curious species are actually reef associated and benthic. They are not pelagic.

Reviewer #2: This study is interesting and could be a valuable contribution in the branch of fish ecology at local and regional scale; however, its contribution is limited in terms of added improvement in the field of methodologies to survey the structure of fish communities, and the manuscript must be written with this in consideration. The study of sources of variation is extensive in fish sampling and most of the elements analyzed here are well known within rules of thumb of sampling design and data capture:

1. Fish communities change in space and time, and this should be considered in sampling designs. 2. Different sampling efforts (number, size, or duration of sampling units) will render different results, the more the better, but the optimum depends on how diverse the community under study is. 3. Different methods of capturing information (video vs visual, static vs transects, camera vs diver) have specific advantages and disadvantages for the capture of abundance and diversity of fish, which need to be considered according to the habitat and question at hand. 4. Divers have an effect on the fish community and different divers will collect different information, and this will depend on their experience in great part, so surveyors need to have an adequate level of knowledge to be surveyors, again, according to the complexity of the system under study.

The study focused on a particular method of diver-based roaming video transects in a nested design with repeated-measures and its results confirm most of these rules of thumb. So, what exactly is this study addressing that has not been done before to make this study unique from previous studies? I believe the key point of the study is that previous studies focused on only a few aspects of the sources of variability while the authors of this paper carried out a more complete inclusion of these by carrying a complex sampling design. However, the benefits of doing this should be well explained. For this the study needs to clearly outline the work that has been done previously regarding sources of bias and error in fish communities’ assessments to clearly identify the knowledge gap that will be addressed. Objectives, questions and/or hypothesis of the study need to be clearly outlined in abstract and introduction, including independent variables (elements of sampling and analyses) and dependent variables (elements of community structure).

I have also made several comments in the manuscript it self. but here I stress some general things. The language and terms used to refer to sampling variability must be clearly defined within the discipline of ecology. The use of terms such as design errors, detection heterogeneity, observer bias, observer effects, random counting and instantaneous fish displacement sounds all confusing, but I think it all can be explained more simply to avoid confusions. There are some statements in the introduction that are not accurate or sufficiently backed up with other studies regarding the use or adequacy of different survey methods.

The study considers abundance and diversity of fish, but did not presented changes in species composition, which is one of the most fundamental factors affected by different survey methods. I suggest including this explicitly.

The discussion needs to avoid repeating results and just aiming to interpret the findings and its significance concisely.

Similarly, the conclusions should focus on the contributions of your study. I would also suggest that given the extensive nesting design and interacting factors, the interpretation and concluding remarks should be stated clearly in the discussion as a table summarizing a set of recommendations for practitioners framed within the specific context of the study (i.e. ecosystem/habitat surveyed). It would be good for example to have carried out surveys in different habitats (need to specify in the manuscript if all transects were done in the same habitat) so the findings could have a greater applicability.

6. PLOS authors have the option to publish the peer review history of their article (what does this mean?). If published, this will include your full peer review and any attached files.

Reviewer #1: No

Reviewer #2: No

---

## [Author Response · Author response to Decision Letter 0]

12 Oct 2021

We thank the reviewers for their helpful comments and interest in the topic. The remarks have contributed to the outcome of the reviewed manuscript and have, in our opinion, increased the quality of its content and structure. In the next section we will clarify how we integrated the comments and suggestions in this reviewed version of the manuscript. 

Reviewer #1: Overview

This paper reports on the accuracy, precision and repeatability of visual transects of shallow water reef fish recorded with underwater video. In particular, the authors aim to address whether there were species-specific observer affects, the importance of observer bias sampling variability, and the importance of transects length number repeats the counting metrics and the way that data was transformed.

As an initial comment the authors do not recognise that the level of transformation that you use is largely dependent on the ecological question that you are asking.

Answer: Indeed, the choice for a transformation is largely dependent on the research question, but that does not alter the fact that it might be interesting to know how different the results would have been if different transformations would have been used. We clarified this by adding following sentence: “Although the choice for a data transformation or dissimilarity index is highly dependent on the research question, researchers might still be interested in their effect on model fit.”

I also note that they are missing some of the fundamental literature including previous similar work and recommend that they actually read the reference below which is fundamental to this field of study.

McCormick, M.I., Choat, J.H. Estimating total abundance of a large temperate-reef fish using visual strip-transects. Mar. Biol. 96, 469–478 (1987). https://doi.org/10.1007/BF00397964

Answer: We have gone through the reference and found indeed many conclusions that were re-established in more recent publications which we already cited. We now also included this reference to support some statements in the manuscript.

I found the paper extremely long and overly complicated. Also, while I recognise that the statistics used are valid, I am not convinced they are the optimal methods for conveying and supporting the authors interpretation presented in the discussion.

Answer: We agree that the manuscript might be quite complex, but we believe that it is important to discuss aspects which are often gone over quite quickly such as the correct partioning of variability among different factors, the limitations of the chosen design, etc. The choice for the different statistical techniques was not made lightly, we believe the chosen methods to be most appropriate.

I’m also concerned that the authors may have unwittingly introduced another source of bias into their design which may well greatly impact their analysis and data collection.

At lines 144 to 145 the authors describe laying out a transact rope and recording 2.5 m either side of the transact. Did they delineate the whole transect by laying out a rectangle or did they lay out a rope down the centre and then the estimated the distance either side of that transect by the image analyst? If they did the latter can the author’s please clearly define how they actually estimated distance. Distance estimates are 1 of the main sources of error that come into fish transects as it can be extremely difficult to consistently estimate distance. As a consequence, an observer may greatly over or underestimate the actual area observed.

Answer: Indeed, both the observers and image analysts were trained to estimate a distance of 2.5 meter on either side of the rope and to decide whether fish had to be included or not. They were trained to estimate the distance but the transects themselves had no delineation of the 2.5 meter borders. These borders had to be estimated on site. It is indeed correct that these estimates will not always be accurate. Therefore we have added following to the manuscript: “It should be noted that, despite their training, observers and image analysts might have over or underestimated the distance from the rope, inflating as such the inter-observer variability and/or sampling variability to some extent.”

See Harvey E, Fletcher D, Shortis MR, Kendrick GA. A comparison of underwater visual distance estimates made by scuba divers and a stereo-video system: implications for underwater visual census of reef fish abundance. Marine and Freshwater Research. 2004 Sep 23;55(6):573-80.

Answer: Thank you for the interesting reference that discusses the issue of distance estimates. We have included it as a reference for the earlier mentioned statement.

The authors also talk about a standard operating procedure developed by the aquatic ecology research group of Ghent University. It would be really useful if this could be deposited as an online appendix so that people could refer to that. Also, how does that align with already published standard operating procedures for doing video transects surveys of fish?

Answer: The methodology description that follows is actually the standard operating procedure we are referring to. We have removed this sentence now as it might actually be confusing for the reader.

I also believe there are some real fundamental errors with this paper in that the authors are trying to count mobile fishes that are quite large and also small fishes such as travel fins and Lenny’s which are habitat associated. Most fish ecologists would use transects of different sizes depending on the mobility and size of the species that they are trying to investigate. Trying to sample them all in one survey is probably not realistic or ecologically valid. I certainly would not be trying to survey fish such as the Galapagos triplefin blenny (max size 9cm and highly territorial) in the same transects that I would be targeting the Amarillo snapper (a species up to 71 cm in length and highly mobile).

Answer: Our main aim is to get an idea of the fish assemblage structure with as little effort as possible. Including more transects of different sizes to assess different species would indeed be more correct, but would also require much more time in the field. Here we want to optimize video transects to assess fish assemblage structures and compare them among sites in a relatively short time. 

It would be great if the authors can make some comment about why they have lumped the species together when most guidance on surveying reef fish would recommend keeping them separate.

Answer: We added following to the text to clarify our choice: “Because we aimed to provide a quick and mobile assessment of the local fish assemblages and limit the sampling effort per site, it was decided to count all fish species \\citep{Bernard2013,Watson2010} and not to count discrete groups of species in transects of different, earlier established, optimal designs \\citep{McClanahan2007,Wilson2018}.”

Overall I think the paper has the potential to be a valid contribution to science, but it needs to be substantially rewritten so that the text and the data presented a simplified. At the moment it is overly and unnecessarily complex. I recommend simplifying the analysis and the presentation of results so that readers can comprehend it and absorb the main takeaway messages. I have included a number of minor comments below.

Answer: We went over the manuscript and simplified many of the sentences and representations of the results. 

Abstract

Line 6. I recommend that you change should to could given that this work (admittedly not for video sampling) is a repeat of past research.

Answer: done

Introduction

Line 56. I recommend that you write this as In the literature.

Answer: done

Line 62. I recommend that the authors put a reference about underwater visual census and some species being scared away and not been detected.

Answer: Three references were added

Lines 59-67. I commend the authors for this distinction which is actually really important.

Answer: Thank you

Line 78 to 80. This statement is actually incorrect. Because of the bait plume from stationary cameras (and I am assuming you are talking about baited remote underwater video systems given you refer to Mike Cappo’s research above) have been demonstrated to be a more effective tool sampling across broad habitats and depth scales than underwater visual census. They tend to sample more species, with greater statistical power than underwater visual census. However, the bias of baited remote underwater video systems are acknowledged and one of the things that you cannot do is get density or biomass estimates per unit area.

Answer: Indeed. The use of baited stationary devices remediates some of the limitations of traditional non-baited stationary devices. We have rewritten this part: Nevertheless, stationary cameras only provide a limited view of the study area as only small patches are monitored, and are therefore of limited use when an extensive survey of a study area has to be done \\citep{Willis2000}. Baited stationary cameras remediate this limitation to some extent by attracting nearby fish, providing a more representative assessment of the local fish assemblages \\citep{Harvey2007}. However, due to its point-based nature, it remains difficult to accurately estimate fish densities and account for habitat heterogeneity \\citep{Andradi-Brown2016,Willis2000}. Because fish species, and in particular reef fish, are often strongly associated with specific micro-habitats \\citep{Chapman2000}, a more mobile approach may often be more appropriate for more cost-effective monitoring of fish densities.

They are also not good for getting counts of some of the smaller and less mobile species. I think you probably want to either reword this sentence or alter it to reflect a true statement.

Similarly, I think in the introduction you need to talk about the types of information that ecologists and managers want to collect and use. You have not talked all about the use of length data which some ecologists would consider to be more fundamentally useful than count data. This aspect should probably be brought into the discussion as well.

Answer: In the introduction we delineate the objective of the manuscript and discuss the value of obtaining insight in the structure and diversity of fish assemblages. Length measurements can indeed be very useful for specific ecological questions, but we don’t believe that mentioning this here would provide much added value, especially since we have been trying to reduce the length of the manuscript. 

Can you provide a reference for line 87 please.

Answer: We provided two references.

Lines 171-174. Can the authors please justify why they have used Max count or min count? One of the great advantages of transects is that you can actually obtain a density estimate. By counting all the fish that you see with in a transact you may increase your variability, but you will also increase data on length frequency. This is a fundamental advantage of doing transects over stationary camera systems. Max count is a term that comes from the baited underwater video literature and is probably inappropriate in this context. Essentially all you are doing is counting all the species and individuals that you see in a transect? I would clarify that and change your figures accordingly.

Answer: We have clarified why we determined both metrics: Fish were counted in such a way that both MaxCount (total number of individuals per species) and MinCount (maximum number of individuals per species in one frame, also referred to as MaxN) could be determined (see section \\ref{Video_analysis}). Since MaxCount was found most appropriate for our video transects (see section \\ref{Variabilityinthestructureoffishassemblages}), which is in line with results from previous studies on video transects \\citep{Mallet2014,Wartenberg2015}, it was used instead of MinCount.

Somewhere in your methods you need to tell the reader how you analysed your video imagery and why used that particular method. At the moment, someone wanting to repeat your work couldn’t do it as that information is not in the main text or the appendices.

Answer: This information is provided in Appendix B. We have expanded the description of the method and included also why it was done as such.

Line 179 to 181. Please make a comment about what the effect of not being able to identify the exact length of the transects might have on the data. It seems like this is probably going to increase a source of error and variability as you may be greatly increasing or decreasing the area surveyed.

Answer: We have clarified what the effect might be of not being able to get exact distances: “Although the inability to provide exact distances will introduce some additional variability, this variability is assumed to be stochastic and limited.”

Line 190. Do you have an example or a reference supporting your thoughts about instantaneous fish displacement? Has anyone else observed this in the scientific literature?

Answer: The number of studies to repeat transects at relatively short time intervals is limited. In most studies transects are not repeated and hence this issue does not occur. However, these studies are also not able to identify the different origins of variability. There are advantages and disadvantages to both approaches, but given our research questions, we believed this approach to be most appropriate. The few studies that do repeat transects don’t go into much detail when describing the impact this might have on the data analysis. We believe this however an important aspect to include. We see this as a gap in current literature as we have not found any suitable references to support this message. A disadvantage of describing this in depth might however be that the text becomes more complex, but we believe it necessary to be included in order to be scientifically sound. 

Figure 3. In the methods I think it would be really useful of the authors stated why they have used the similarity values to represent their transects and not looked at the cumulative number of species or individuals and also presented the means. The dissimilarity values are a valid way of presenting this, but an unusual way of presenting this data. I’m not convinced that this the optimal way of presenting it.

Answer: First, in this manuscript we mainly focussed on the ability of video transects to be used for assessments of fish assemblage structure. We also included assessments of diversity and in the appendix we included some species accumulation curves for different transect lengths which are briefly referred to in the discussion to support a statement. However the focus is on fish assemblage structure. Second, this design error we are interested in is defined in the methods section as: “An important design error may arise from the choice of transect length. If transect lengths are too short, many observations may contain no species at all and there may also be pairs of observations having no or only few species in common, leading to uninformative dissimilarity values and questionable multivariate analyses \\citep{Clarke2006}”. We therefore clearly define this design error within the context of multivariate analyses of fish assemblage structure, justifying the use of frequencies of dissimilarity values.

Lines 429 - 442. Rather than using the components of variation have the authors considered using a distance based linear model which might be easier for the reader to understand?

Answer: Given that all the variables that we considered were categorical, a PERMANOVA was found more suitable and more straightforward to interpret than a DISTLM (Anderson et al., 2008). 

Line 464. Please change the use of species density as if you are using min count or Max count you’re not counting all the individuals of a species. Again, you have used Max count inappropriately.

Answer: We define MaxCount as the total number of individuals per species, which means that all fish are counted. Species density is the number of species per unit of surface. That would be the same for MinCount or MaxCount. Only MaxCount is used throughout the analyses. 

Line 533 - 534. Please note that both the species that you list as pelagic curious species are actually reef associated and benthic. They are not pelagic.

Answer: We corrected this.

Reviewer #2: Overview

 This study is interesting and could be a valuable contribution in the branch of fish ecology at local and regional scale; however, its contribution is limited in terms of added improvement in the field of methodologies to survey the structure of fish communities, and the manuscript must be written with this in consideration. The study of sources of variation is extensive in fish sampling and most of the elements analyzed here are well known within rules of thumb of sampling design and data capture:

1. Fish communities change in space and time, and this should be considered in sampling designs. 2. Different sampling efforts (number, size, or duration of sampling units) will render different results, the more the better, but the optimum depends on how diverse the community under study is. 3. Different methods of capturing information (video vs visual, static vs transects, camera vs diver) have specific advantages and disadvantages for the capture of abundance and diversity of fish, which need to be considered according to the habitat and question at hand. 4. Divers have an effect on the fish community and different divers will collect different information, and this will depend on their experience in great part, so surveyors need to have an adequate level of knowledge to be surveyors, again, according to the complexity of the system under study.

The study focused on a particular method of diver-based roaming video transects in a nested design with repeated-measures and its results confirm most of these rules of thumb. So, what exactly is this study addressing that has not been done before to make this study unique from previous studies? I believe the key point of the study is that previous studies focused on only a few aspects of the sources of variability while the authors of this paper carried out a more complete inclusion of these by carrying a complex sampling design. However, the benefits of doing this should be well explained. For this the study needs to clearly outline the work that has been done previously regarding sources of bias and error in fish communities’ assessments to clearly identify the knowledge gap that will be addressed. Objectives, questions and/or hypothesis of the study need to be clearly outlined in abstract and introduction, including independent variables (elements of sampling and analyses) and dependent variables (elements of community structure).

Answer: We believe that the motivation for this specific design has been described in the introduction from line 107 to line 135. Regarding the objectives, these are given from line 136 to 140 at the end of the introduction. We did not want to go too much into the methodological aspects of the objectives at this point because we hadn’t formulated the aspects of the sampling design and technique yet. This would be of little added value for the reader and might even be confusing. 

I have also made several comments in the manuscript itself. but here I stress some general things. The language and terms used to refer to sampling variability must be clearly defined within the discipline of ecology. The use of terms such as design errors, detection heterogeneity, observer bias, observer effects, random counting and instantaneous fish displacement sounds all confusing, but I think it all can be explained more simply to avoid confusions. There are some statements in the introduction that are not accurate or sufficiently backed up with other studies regarding the use or adequacy of different survey methods.

Answer: In this manuscript we have relied on the terminology and definitions used in literature. We provided some additional clarifications for some terms throughout the text. 

In the abstract only the most essential information can be presented. Therefore, we could not elaborate on all the aspects and results of the study in the abstract. 

Comments in the pdf we have answered within the pdf: PONE-D-21-06060_errors and variability in video transects_SB.pdf 

The study considers abundance and diversity of fish, but did not presented changes in species composition, which is one of the most fundamental factors affected by different survey methods. I suggest including this explicitly.

Answer: We provide assessments of the fish assemblage structure which takes into account both species composition and abundance. We also compared the repercussions of choosing presence/absence data versus abundance data, providing an additional assessment of species composition without considering abundance. 

The discussion needs to avoid repeating results and just aiming to interpret the findings and its significance concisely.

Answer: We have improved the brevity of sentences in the discussion by focussing more on the interpretations and less on the results. 

Similarly, the conclusions should focus on the contributions of your study. I would also suggest that given the extensive nesting design and interacting factors, the interpretation and concluding remarks should be stated clearly in the discussion as a table summarizing a set of recommendations for practitioners framed within the specific context of the study (i.e. ecosystem/habitat surveyed). It would be good for example to have carried out surveys in different habitats (need to specify in the manuscript if all transects were done in the same habitat) so the findings could have a greater applicability.

Answer: We prefer to keep these recommendations in the discussion as it allows to provide more context. If we would do this in the conclusion, the conclusion would be too long and not of much added value. 

References

Anderson, M.J., Gorley, R.N., Clarke, K.R., 2008. PERMANOVA+ for PRIMER: Guide to Software and Statistical Methods, PRIMER-E Ltd., Plymouth, UK.

---

## [Decision Letter · Decision Letter 1]

6 Dec 2021

PONE-D-21-06060R1

Sampling errors and variability in video transects for assessment of reef fish assemblage structure and diversity

PLOS ONE

Dear Dr. Bruneel,

Thank you for submitting your manuscript to PLOS ONE. After careful consideration, we feel that it has merit but does not fully meet PLOS ONE’s publication criteria as it currently stands. Therefore, we invite you to submit a revised version of the manuscript that addresses the points raised during the review process.

The two reviewers concur that the revision does not address (or does not fully address) many of the comments and issues raised by the reviewers of the original manuscript. They have a number of concerns with this revised version, and offer several comments and suggestions to improve it. If you will address these individually and in detail, we look forward to receiving a revised text.

We look forward to receiving your revised manuscript.

Kind regards,

Antonio Medina Guerrero, Ph.D.

Academic Editor

PLOS ONE

Reviewers' comments:

Reviewer's Responses to Questions

**Comments to the Author**

1. If the authors have adequately addressed your comments raised in a previous round of review and you feel that this manuscript is now acceptable for publication, you may indicate that here to bypass the “Comments to the Author” section, enter your conflict of interest statement in the “Confidential to Editor” section, and submit your "Accept" recommendation.

Reviewer #2: (No Response)

Reviewer #3: (No Response)

2. Is the manuscript technically sound, and do the data support the conclusions?

Reviewer #2: Yes

Reviewer #3: Partly

3. Has the statistical analysis been performed appropriately and rigorously? 

Reviewer #2: Yes

Reviewer #3: Yes

4. Have the authors made all data underlying the findings in their manuscript fully available?

Reviewer #2: No

Reviewer #3: Yes

5. Is the manuscript presented in an intelligible fashion and written in standard English?

Reviewer #2: Yes

Reviewer #3: Yes

6. Review Comments to the Author

Reviewer #2: The authors have made some improvements in the text (some are not highlighted in the resubmitted document), such as including more references and adjusting word choice and clarifications. Unfortunately, the authors have not addressed other comments, and thus, there are still some inaccuracies that need attention before acceptance for publication. Please see comments below with reference to line numbers of the new submitted version.

Abstract (no numbered lines)

Conclusions/significance can be improved. "The results confirm the suitability of the technique to study reef fish assemblages," This was not the objective of the study, so this was not proved.

Introduction

Lines 33-67. Static videos (RUV) are not adequate for extensive sampling areas but baited videos can improve this. These ideas are incorrect. RUV can be designed to cover extensive areas. Western and Eastern Australia is an example of this. The plume-effect of baited videos is actually a problem, since it can attract fish species from Kilometres away that normally would not be in the sampling location, modifying natural fish assemblages and habitat associations. Also, it is well known that fish densities and habitat heterogeneity can be accounted for in video analyses when designed appropriately (e.g. benthobox). Authors need to be careful with these kind of statements. I recommend to modify accordingly.

Importantly, there has not been an improvement regarding the background information in the introduction to layout the knowledge gap being addressed by the study (Lines 52-65). Similarly, the explanation of the objectives remained unchanged. Thus, as it currently stands, the novel contribution of the study is not clear. Authors need to make this more explicit. This highlights the importance of the study more and also helps the readership to understand what the study is about easily. Aiming to evaluate “how important” is something, is not scientifically clear. The answer to this kind of questions would be “not important, important, mildly important, very important, etc" but we want quantitative answers. So it must be said exactly what you will test.

L 77. “Counting metric, data type and transformation” is referred in the third question/objective but this has not been mentioned at all in the background. If its in the objectives it must be explained why beforehand.

Methods

Table 1 (glossary) remains without legend or reference in the text.

L83. Habitat description is missing (was all the sampling done in the same type of habitat?).

L86. I cannot see attachments with supplementary materials (e.g. Fig. B1 map).

L98. Why did you choose to survey the transect six consecutive times (why not less or more)?

L102. How widely used is this technique (s-transect) around the world?

L110. Time between surveys of different observers was 1 minute, which according to other studies is too short to avoid effects between observers, you should account/discuss this (Reef fish communities are spooked by scuba surveys... PeerJ 6, e4886 (2018)).

L122. Why split the videos as 0-50, 5-45, 10-40, etc. and not 0-50, 0-40, 0-30m etc.?

L125. Provide estimated times-distances (line 125)

The suggestion to include explicit analyses of species composition was also disregarded. Although indeed Diversity indices include composition, this does not tell you which taxa accounts for the differences, which is of high interest to practitioners. This is discussed briefly in the discussion for a few species, but its explicit inclusion in the analyses and presentation would add good value to the study.

Discussion

The concluding statement has not been improved. The current text just says what is already known: “methodological parameters, environmental conditions, and ecological knowledge, e.g. species-specific traits affecting detectability, were found important for video transects and should be accounted for to properly assess the ecological state and the effect of management practices on the environment” but what exactly was discovered in the study that is new? (refer to the hypotheses and objectives).

Reviewer #3: Review of revised manuscript

Title: Sampling errors and variability in video transects for assessment of reef fish assemblage structure and diversity.

Comments to the Author(s):

The manuscript was reviewed and I am in agreement with regard to the comments/corrections suggested by the reviewers. The manuscript was revised to address many of the comments suggested by the reviewers and the authors gave a response for each comment or question. However, some of the responses were not completely addressed or could be completely addressed without repeating the analyses or data collection. For example, counting or lumping all species together with no separation of species by behavior or life stage is an issue which cannot be addressed without reorganizing and re-analyzing the data. The authors did offer responses and added statements to try and address in the revision but the issue is not fixed.

The following are specific comments that require further attention:

Not determining the exact length of transects may have introduced additional variability but this variability isn't measured by the authors or quantified. They suggest it can be assumed to be stochastic and limited. Can it be measured or is assuming it to be stochastic and limited adequate? It may or may not be adequate but a better justification should be given for this point and others that were questioned by the reviewers.

As reviewer 2 pointed out, the Introduction should be written to emphasize the importance of this study in comparison to others and its contribution to the existing literature. I don't believe that the authors response to address this comment is adequate. The foundation for the methods to be use should be established in the Introduction along with clearly stated objectives.

Both reviewers suggested making the objectives more clear in the Introduction section. The authors mentioned that the objectives are given from line 107 to 135. However, the Introduction ends with line 78. There are 3 questions listed at the end of this section but those are not objectives. If the objectives are in fact given at the end, I cannot find them. The study objectives should be stated clearly at the end of the Introduction which they are not. I also agree with reviewers that the Introduction needs to include the a description of the type of data that is important for ecologists and managers and how the study will address ways to collect and analyze the desired types of data.

The choice of the data transformation was stated by the authors as being highly dependent on the research question(s) of the study. Though readers may be interested in how different transformations may affect the results of analyses, the authors didn't really address the how the data transformation addresses their questions or in other words, what the point of using different data transformation was for the conclusions of the current study. There should be more to the answer than just stating that researchers might be interested in. State the importance of determining this result to researchers in the Introduction.

The choice of statistics does make the manuscript rather complicated but justify why you are going to used these methods this in the Introduction. I understand what the authors were trying to do but its not explained completely within the manuscript so that the readers will understand what the objectives were and why specific analyses were done.

Lastly, the following two comments given by reviewer 2 were not adequately addressed.

1. The study considers abundance and diversity of fish, but did not presented changes in species

composition, which is one of the most fundamental factors affected by different survey methods. I

suggest including this explicitly.

The authors did not mention if they will add anything to the manuscript regarding changes in species composition. If not, why ?

2. Similarly, the conclusions should focus on the contributions of your study. I would also suggest that

given the extensive nesting design and interacting factors, the interpretation and concluding remarks

should be stated clearly in the discussion as a table summarizing a set of recommendations for

practitioners framed within the specific context of the study (i.e. ecosystem/habitat surveyed). It

would be good for example to have carried out surveys in different habitats (need to specify in the

manuscript if all transects were done in the same habitat) so the findings could have a greater

applicability.

The authors did not add the specific recommendations to the revision nor did they give any indication if they will or will not add them and why.

Just a note, MaxN and MinCount are not density estimates. There is no unit of area included in their calculation. These can be considered abundance metrics but not density metrics.

The following are specific comment/corrections for the text:

Abstract:

Reviewer #1 recommended changing "should" in line 6 which was done by the authors.

Introduction:

Line 5:

What environmental and ecological changes are referred to here? The prior sentence doesn't quite reference changes just physical and chemical water conditions that are known to be affected by anthropogenic pressures. What are "these changes?" Remove "these" from line 4.

Line 34:

Remove the comma and add "thereby" before "providing a more...."

Line 35:

"Baited stationary cameras" is written in the prior sentence but the authors refer to the camera systems as "its" in line 35. Replace "its" with something like "However, due to the point-based nature of these camera systems, it remains difficult....."

Line 74 to 78: The questions seem to just be listed here with little connection to the paragraph above. Why not change the wording to "Although repetitions of transects are not common in the literature, they were necessary here to provide answers to the following research questions, which are of interest for both non-repetitive and repetitive studies:"

Discussion:

Line 443: Remove "two curious species." This statement makes the assumption that the fish were in fact curious which is something that cannot be proven.

Line 446: Replace curious with "non-scared" fish. Being attracted to the divers doesn't mean the fish are actually curious.

Line 462 to line 469: Could the temporal auto-correlation, fish displacement, and observer biases be avoided or greatly reduced by sampling transects over a series of days not one after another on the same day? The sampling events would still be repeats.

7. PLOS authors have the option to publish the peer review history of their article (what does this mean?). If published, this will include your full peer review and any attached files.

Reviewer #2: No

Reviewer #3: No

---

## [Author Response · Author response to Decision Letter 1]

1 Jun 2022

Rebuttal letter

We thank the reviewers for their helpful comments and interest in the topic. The remarks have contributed to the outcome of the reviewed manuscript and have, in our opinion, increased the quality of its content and structure. In the next section we will clarify how we integrated the comments and suggestions in this reviewed version of the manuscript. 

Reviewer #1:

recommended changing "should" in line 6 which was done by the authors.

Introduction:

Line 5:

What environmental and ecological changes are referred to here? The prior sentence doesn't quite reference changes just physical and chemical water conditions that are known to be affected by anthropogenic pressures. What are "these changes?" Remove "these" from line 4.

Answer: “these” was removed.

Line 34:

Remove the comma and add "thereby" before "providing a more...." 

Answer: comma removed and “thereby” added. 

Line 35:

 "Baited stationary cameras" is written in the prior sentence but the authors refer to the camera systems as "its" in line 35. Replace "its" with something like "However, due to the point-based nature of these camera systems, it remains difficult....."

Answer: We corrected it as such. 

Line 74 to 78: The questions seem to just be listed here with little connection to the paragraph above. Why not change the wording to "Although repetitions of transects are not common in the literature, they were necessary here to provide answers to the following research questions, which are of interest for both non-repetitive and repetitive studies:" 

Answer: We corrected it as such. 

Discussion:

Line 443: Remove "two curious species." This statement makes the assumption that the fish were in fact curious which is something that cannot be proven.

Answer: We have rewritten it as “two species which are known to not be frightened easily” and now also refer to literature. 

Line 446: Replace curious with "non-scared" fish. Being attracted to the divers doesn't mean the fish are actually curious.

Answer: We corrected it as such. 

Line 462 to line 469: Could the temporal auto-correlation, fish displacement, and observer biases be avoided or greatly reduced by sampling transects over a series of days not one after another on the same day? The sampling events would still be repeats.

Answer: The issue there might be that the time between observations becomes so large that the assemblages are completely different, not because of issues related to the observation method but simply because of the movement of animals. The question is then whether two observations in two subsequent days can be considered repeats or not. 

Reviewer #2: 

The authors have made some improvements in the text (some are not highlighted in the resubmitted document), such as including more references and adjusting word choice and clarifications. Unfortunately, the authors have not addressed other comments, and thus, there are still some inaccuracies that need attention before acceptance for publication. Please see comments below with reference to line numbers of the new submitted version.

Abstract (no numbered lines)

Conclusions/significance can be improved. "The results confirm the suitability of the technique to study reef fish assemblages," This was not the objective of the study, so this was not proved.

Answer: We included this now in the explicit description of the aim of the study: L53-55: ‘The aim of this study was therefore to provide a detailed assessment of the observer effects, observer bias, sampling variability, sampling design and overall suitability of video transects to study reef fish assemblages.’

Introduction

Lines 33-67. Static videos (RUV) are not adequate for extensive sampling areas but baited videos can improve this. These ideas are incorrect. RUV can be designed to cover extensive areas. Western and Eastern Australia is an example of this. The plume-effect of baited videos is actually a problem, since it can attract fish species from kilometers away that normally would not be in the sampling location, modifying natural fish assemblages and habitat associations. Also, it is well known that fish densities and habitat heterogeneity can be accounted for in video analyses when designed appropriately (e.g. benthobox). Authors need to be careful with these kind of statements. I recommend to modify accordingly.

Answer: We rephrased this: L31-38: “Nevertheless, stationary cameras often only provide a limited view of the study area as only small patches are monitored, and are therefore of limited use when a rapid, yet spatially extensive survey of a study area has to be done \\cite{Willis2000}. Baited stationary cameras remediate this limitation by attracting nearby fish thereby providing a more representative assessment of the local fish assemblages \\cite{Harvey2007}. However, due to the point-based nature of these camera-systems, it remains difficult to accurately account for habitat heterogeneity and assess habitat preferences (especially of less mobile species) \\cite{Andradi-Brown2016,Willis2000}.”

Importantly, there has not been an improvement regarding the background information in the introduction to layout the knowledge gap being addressed by the study (Lines 52-65). Similarly, the explanation of the objectives remained unchanged. Thus, as it currently stands, the novel contribution of the study is not clear. Authors need to make this more explicit. This highlights the importance of the study more and also helps the readership to understand what the study is about easily. Aiming to evaluate “how important” is something, is not scientifically clear. The answer to this kind of questions would be “not important, important, mildly important, very important, etc" but we want quantitative answers. So it must be said exactly what you will test.

Answer: We described the aim of the study early on in the study now: L53-55: “The aim of this study was therefore to provide a detailed assessment of the observer effects, observer bias, sampling variability, sampling design and overall suitability of video transects to study reef fish assemblages.” The need to address this aim is also given in the paragraph before, by highlighting the limited number of studies focusing on only several of the methodological aspects that are targeted here. We have also added the following to highlight the need of this kind of research: L46-50: “Although studies comparing different video and other visual census techniques are becoming more frequent \\cite{Andradi-Brown2016,Langlois2010,Tessier2013,Watson2005,Watson2010,Wilson2018}, few have focused on the methodological aspects of video transects, such as observer bias, design errors, sampling variability, counting metrics, data types and data transformations \\cite{Bernard2013,Goetze2019}.”

L 77. “Counting metric, data type and transformation” is referred in the third question/objective but this has not been mentioned at all in the background. If its in the objectives it must be explained why beforehand.

Answer: We have added this to the text: L46-55:“Although studies comparing different video and other visual census techniques are becoming more frequent \\cite{Andradi-Brown2016,Langlois2010,Tessier2013,Watson2005,Watson2010,Wilson2018}, few have focused on the methodological aspects of video transects, such as observer bias, design errors, sampling variability, counting metrics, data types and data transformations \\cite{Bernard2013,Goetze2019}. Video transects are often considered a simple extension of visual census transects, although the ability to store and standardize video observations (see \\nameref{ap:add:applic}) justifies a specific methodological assessment of the technique. The aim of this study was therefore to provide a detailed assessment of the aforementioned methodological aspects and overall suitability of video transects to study reef fish assemblages.” We also included a more detailed description of the meaning/interpretation of data transformations: L274-279: ‘In addition, the considered transformation of abundance data to presence-absence data entails a broader discussion regarding the way data should be collected (i.e. identification and counting versus only identification respectively).’ And L591-595: ‘Although neglecting the abundance of a species scored worst in terms of goodness-of-fit, the difference between MinCount and presence-absence data was rather small. Therefore, in this specific case, if one wanted to reduce the time required for video analysis, the presence-absence instead of the abundance data could have been used, without much information being lost.’ We do think however that the issue of data transformation does not fit well in the introduction as it would negatively affect the structure. If we would discuss all methodological aspects in the introduction in detail it would negatively affect the clarity of the introduction. We do, however, now also mention it explicitly in the aim of the study in the introduction: L56-58.

Methods

Table 1 (glossary) remains without legend or reference in the text.

Answer: We have added a caption and referred to the glossary when first using the terms that are explained in the glossary. 

L83. Habitat description is missing (was all the sampling done in the same type of habitat?).

Answer: The habitat description was added: L91-93: “At ten locations, fixed transects were used to assess reef fish assemblages in rocky habitats close to the coast.”

L86. I cannot see attachments with supplementary materials (e.g. Fig. B1 map).

Answer: These attachments were uploaded, but are perhaps not available to the reviewers for some reason. 

L98. Why did you choose to survey the transect six consecutive times (why not less or more)?

Answer: We have now added this information: L106-111: ‘To quantify observer bias and sampling variability, a sufficiently high number of repeats was required, yet the maximal number of observations per day was limited by light availability and fatigue of the observers. Balancing these requirements resulted in each of the transects being recorded six consecutive times with single GoPro cameras (GoPro Hero 5 Black, 1080p, 60fps, wide FOV) by three different observers equipped with a mask and snorkel.’

L102. How widely used is this technique (s-transect) around the world?

Answer: It is widely used because of its mentioned advantages. We added this L113: “The observers covered the transects in a browsing fashion, similar to the widely used S-type transects introduced by \\cite{Pelletier2011}.”

Answer

L110. Time between surveys of different observers was 1 minute, which according to other studies is too short to avoid effects between observers, you should account/discuss this (Reef fish communities are spooked by scuba surveys... PeerJ 6, e4886 (2018)).

That is one of the things we assess here: The observer effects. The effects of the presence of the observer on fish communities. Hence we do discuss it in depth. We have made this more clear in the methods section: L121-122: “The time between successive observations was at least one minute. The duration between observations was most often too short to consider observations as independent \\cite{Emslie2018ReefRecover}. These dependencies between observations are described as observer effects, which will be discussed further.”

L122. Why split the videos as 0-50, 5-45, 10-40, etc. and not 0-50, 0-40, 0-30m etc.?

To limit disturbances, observers would cover a transect in both directions. We have also added this in the text now. L124-125: “Observers covered the transects back to back, so three times in one direction and three times in the other direction.” Unfortunately, it was not recorded in which direction an observer started the transect and it was very difficult to deduce this from the videos. Which means that the section 0-10 might actually have been 40-50. This is why we chose to work with the sections 5-45 and so on. 

L125. Provide estimated times-distances (line 125) 

Answer: this remark was not entirely clear to us. 

The suggestion to include explicit analyses of species composition was also disregarded. Although indeed Diversity indices include composition, this does not tell you which taxa accounts for the differences, which is of high interest to practitioners. This is discussed briefly in the discussion for a few species, but its explicit inclusion in the analyses and presentation would add good value to the study.

Answer: We have now also added a description of the most important species affecting the fish assemblage structure differences between both islands. L400-409: “Given the pronounced difference of the fish assemblage structure between islands, the first PCO axis of a PCO analysis of the full data was a good proxy to distinguish the fish assemblage structures of both islands. Using the correlations of the full data with this first PCO axis, the species that are most important to explain this difference between islands, could be identified. The most important species to explain differences in fish assemblage structure between the islands included, for Santa Cruz, the Bullseye puffer fish (\\textit{Sphoeroides annulatus}; \\textit{r} = $-$0.62), Yellowtail damselfish (\\textit{Microspathodon bairdii}; \\textit{r} = $-$0.83) and Pacific spotfin mojarra (\\textit{Eucinostomus dowii}; \\textit{r} = $-$0.64), and, for Floreana, the Bravo clinid (\\textit{Gobioclinus dendriticus}; \\textit{r} = 0.56), Galapagos ringtail damselfish (\\textit{Stegastes beebei}; \\mbox{\\textit{r} = 0.83}) and Chameleon wrasse (\\textit{Halichoeres dispilus}; \\textit{r} = 0.87).”

Discussion

The concluding statement has not been improved. The current text just says what is already known: “methodological parameters, environmental conditions, and ecological knowledge, e.g. species-specific traits affecting detectability, were found important for video transects and should be accounted for to properly assess the ecological state and the effect of management practices on the environment” but what exactly was discovered in the study that is new? (refer to the hypotheses and objectives).

Answer: We have adapted the conclusion accordingly and provide clear answers to the hypotheses and objectives of the study: L535-559: “Deciding on the distribution of the sampling effort among different methodological aspects is an important step for any study. In this study we described how researchers should take into account a broad range of potentially conflicting theoretical and practical aspects when setting up an experimental design for video transects. For most species, the presence of an observer has a significant effect on fish behavior which introduces severe dependence between subsequent observations. These observer effects seem however negligible when the overall fish assemblage structure is assessed rather than the counts of individual species. Similarly, the overall sampling variability (consisting of observer effects, instantaneous fish displacement and random counting errors) can for some individual species be very high (up to 100 \\%), while for community assessments it remains relatively low (17-26\\%). The errors associated with the perception of the observers themselves on the other hand, i.e. observer bias, is negligible for both community and species assessments (mostly below 1\\%). Longer transects have a much lower sampling variability (33\\% lower for community assessments), yet to increase precision of repeated observations, more repeats seem more effective than longer transects. For transects, the MaxCount is preferred over MinCount as the former seems to yield a lower sampling variability even though the video processing time is the same. Using counts instead of presence/absence data is also preferred, as the sampling variability of the observations of the former is somewhat lower. However, if the amount of time available for video processing is limited, presence/absence observations still provide observations of reasonably low sampling variability. In summary, sampling variability, methodological parameters, environmental conditions, and ecological knowledge, e.g. species-specific traits affecting detectability, are important for video transects and should be accounted for to properly assess the ecological state and the effect of management practices on the environment.” 

 

Reviewer #3: 

Comments to the Author(s):

The manuscript was reviewed and I am in agreement with regard to the comments/corrections suggested by the reviewers. The manuscript was revised to address many of the comments suggested by the reviewers and the authors gave a response for each comment or question. However, some of the responses were not completely addressed or could be completely addressed without repeating the analyses or data collection. For example, counting or lumping all species together with no separation of species by behavior or life stage is an issue which cannot be addressed without reorganizing and re-analyzing the data. The authors did offer responses and added statements to try and address in the revision but the issue is not fixed. 

Answer: We have added in Table G.2, a literature-based score of the reaction to divers of the different species. “The reaction-to-observer provides a literature-based score from 1, for seemingly shy and easily frightened species, to 6, for seemingly curious species \\citep{Humann2003ReefGalapagos}.” In the methods section we mention that “A literature-based score of reaction-to-observer was compared with the results of these models \\cite{Humann2003ReefGalapagos}.” And this is described in the results section as: L326-329: “For 9 out of 11 species with significant observer effects, the literature-based score of the reaction-to-observer corresponded with the observed fish behavior towards observers (Table G.2). There was a discrepancy in the observed effects and literature-based score for the Amarillo snapper and Yellowtail damselfish.”

The following are specific comments that require further attention:

Not determining the exact length of transects may have introduced additional variability but this variability isn't measured by the authors or quantified. They suggest it can be assumed to be stochastic and limited. Can it be measured or is assuming it to be stochastic and limited adequate? It may or may not be adequate but a better justification should be given for this point and others that were questioned by the reviewers.

Answer: It is not possible to quantify the additional variability that is introduced by not exactly knowing how long the transects are, but we provided more information to support our assumptions: L141-143: “Although the inability to provide exact distances will introduce some additional variability that cannot be quantified, we assumed this variability to be stochastic and limited as observers were specifically trained to maintain a constant swimming speed.”

As reviewer 2 pointed out, the Introduction should be written to emphasize the importance of this study in comparison to others and its contribution to the existing literature. I don't believe that the authors response to address this comment is adequate. The foundation for the methods to be use should be established in the Introduction along with clearly stated objectives.

Answer: We have rewritten the last part of the introduction. Now the aim of the study is written down more explicitly, more early on in the manuscript and within the context of existing gaps in literature. L46-55: “Although studies comparing different video and other visual census techniques are becoming more frequent \\cite{Andradi-Brown2016,Langlois2010,Tessier2013,Watson2005,Watson2010,Wilson2018}, few have focused on the methodological aspects of video transects, such as observer bias (Table \\ref{tab:gloss}), design errors (Table \\ref{tab:gloss}), sampling variability, counting metrics, data types and data transformations \\cite{Bernard2013,Goetze2019}. Video transects are often considered a simple extension of visual census transects, although the ability to store and standardize video observations (see \\nameref{ap:add:applic}) justifies a specific methodological assessment of the technique. The aim of this study was therefore to provide a detailed assessment of the aforementioned methodological aspects and overall suitability of video transects to study reef fish assemblages.”

Both reviewers suggested making the objectives more clear in the Introduction section. The authors mentioned that the objectives are given from line 107 to 135. However, the Introduction ends with line 78. There are 3 questions listed at the end of this section but those are not objectives. If the objectives are in fact given at the end, I cannot find them. The study objectives should be stated clearly at the end of the Introduction which they are not. I also agree with reviewers that the Introduction needs to include the a description of the type of data that is important for ecologists and managers and how the study will address ways to collect and analyze the desired types of data. 

Answer: We have written down the objectives as research questions. We now specify the general aim of the study and break it down in three research questions, which are introduced more clearly now in the introduction. 

The choice of the data transformation was stated by the authors as being highly dependent on the research question(s) of the study. Though readers may be interested in how different transformations may affect the results of analyses, the authors didn't really address the how the data transformation addresses their questions or in other words, what the point of using different data transformation was for the conclusions of the current study. There should be more to the answer than just stating that researchers might be interested in. State the importance of determining this result to researchers in the Introduction. 

Answer: We included a more detailed description of the meaning/interpretation of data transformations: L274-279: ‘In addition, the considered transformation of abundance data to presence-absence data entails a broader discussion regarding the way data should be collected (i.e. identification and counting versus only identification respectively).’ And L591-595: ‘Although neglecting the abundance of a species scored worst in terms of goodness-of-fit, the difference between MinCount and presence-absence data was rather small. Therefore, in this specific case, if one wanted to reduce the time required for video analysis, the presence-absence instead of the abundance data could have been used, without much information being lost.’ We do think however that the issue of data transformation does not fit well in the introduction as it would negatively affect the structure. If we would discuss all methodological aspects in the introduction in detail it would negatively affect the clarity of the introduction. We do, however, now also mention it explicitly in the aim of the study in the introduction. 

The choice of statistics does make the manuscript rather complicated but justify why you are going to use these methods this in the Introduction. I understand what the authors were trying to do but it’s not explained completely within the manuscript so that the readers will understand what the objectives were and why specific analyses were done.

Answer: Discussing this in the introduction in detail would make the introduction very hard to follow and draw the attention away from the other aspects that need to be addressed in the introduction. We added following in the introduction to highlight the need for the specific methods and also refer the reader to the part of the manuscript at which more information is provided: L78-82: “The innovative, yet complex design of this study can provide new insights in the methodology of observation techniques, yet requires multiple advanced statistical methods. An overview of these methods and reasoning behind their use is provided in section \\ref{Data_analysis}.”

Lastly, the following two comments given by reviewer 2 were not adequately addressed.

1. The study considers abundance and diversity of fish, but did not presented changes in species 

composition, which is one of the most fundamental factors affected by different survey methods. I 

suggest including this explicitly. The authors did not mention if they will add anything to the manuscript regarding changes in species composition. If not, why ?

Answer: We made this more clear in the text by adding: L276-279: ‘Finally, following the predominant terminology used in literature, the multivariate analyses of abundance and presence/absence data will yield insights in the structure and species composition of fish assemblages, respectively.’ Hence the distinction between fish assemblage structures and species composition can be brought back to the use of abundance versus presence/absence data. The repercussions of different types of data (abundance versus presence/absence) is discussed in the discussion. 

2. Similarly, the conclusions should focus on the contributions of your study. I would also suggest that 

given the extensive nesting design and interacting factors, the interpretation and concluding remarks 

should be stated clearly in the discussion as a table summarizing a set of recommendations for 

practitioners framed within the specific context of the study (i.e. ecosystem/habitat surveyed). It 

would be good for example to have carried out surveys in different habitats (need to specify in the 

manuscript if all transects were done in the same habitat) so the findings could have a greater 

applicability. The authors did not add the specific recommendations to the revision nor did they give any indication if they will or will not add them and why.

Answers: it was added that all transects were laid out in rocky habitats (L91-92). Specific habitat characteristics were not available, but habitats were very similar as rocky habitats were targeted for placement of the transects. Instead of adding a table we have extended the conclusions with the most important results of the manuscript. Most of the recommendations need to be discussed in the context of the aims, area, etc. which would take too much space in a table. Recommendations in a table without this context would lead to potential misinterpretation and we want to avoid this. 

Just a note, MaxN and MinCount are not density estimates. There is no unit of area included in their calculation. These can be considered abundance metrics but not density metrics.

Indeed. They are not being referred to as density metrics in the manuscript. All mention of fish densities have been removed and/or replaced by ‘fish abundance(s)’.

---

## [Decision Letter · Decision Letter 2]

23 Jun 2022

Sampling errors and variability in video transects for assessment of reef fish assemblage structure and diversity

PONE-D-21-06060R2

Dear Dr. Bruneel,

We’re pleased to inform you that your manuscript has been judged scientifically suitable for publication and will be formally accepted for publication once it meets all outstanding technical requirements.

Kind regards,

Antonio Medina Guerrero, Ph.D.

Academic Editor

PLOS ONE

Additional Editor Comments (optional):

Reviewers' comments:

Reviewer's Responses to Questions

**Comments to the Author**

1. If the authors have adequately addressed your comments raised in a previous round of review and you feel that this manuscript is now acceptable for publication, you may indicate that here to bypass the “Comments to the Author” section, enter your conflict of interest statement in the “Confidential to Editor” section, and submit your "Accept" recommendation.

Reviewer #3: All comments have been addressed

2. Is the manuscript technically sound, and do the data support the conclusions?

Reviewer #3: Yes

3. Has the statistical analysis been performed appropriately and rigorously? 

Reviewer #3: Yes

4. Have the authors made all data underlying the findings in their manuscript fully available?

Reviewer #3: Yes

5. Is the manuscript presented in an intelligible fashion and written in standard English?

Reviewer #3: Yes

6. Review Comments to the Author

Reviewer #3: Reviewer #3:

Comments to the Author(s):

The revised manuscript is an improvement over the previous version. The authors made an effort to address all the questions/comments that I addressed. Overall, I think it is suitable for acceptance for publication if the editor agrees.

The manuscript was reviewed and I am in agreement with regard to the comments/corrections suggested by the reviewers. The manuscript was revised to address many of the comments suggested by the reviewers and the authors gave a response for each comment or question. However, some of the responses were not completely addressed or could be completely addressed without repeating the analyses or data collection. For example, counting or lumping all species together with no separation of species by behavior or life stage is an issue which cannot be addressed without reorganizing and re-analyzing the data. The authors did offer responses and added statements to try and address in the revision but the issue is not fixed.

Answer: We have added in Table G.2, a literature-based score of the reaction to divers of the different species. “The reaction-to-observer provides a literature-based score from 1, for seemingly shy and easily frightened species, to 6, for seemingly curious species \\citep{Humann2003ReefGalapagos}.” In the methods section we mention that “A literature-based score of reaction-to-observer was compared with the results of these models \\cite{Humann2003ReefGalapagos}.” And this is described in the results section as: L326-329: “For 9 out of 11 species with significant observer effects, the literature-based score of the reaction-to-observer corresponded with the observed fish behavior towards observers (Table G.2). There was a discrepancy in the observed effects and literature-based score for the Amarillo snapper and Yellowtail damselfish.”

Reviewer 3 response:

The authors are not able to separate the species by behavior or life stage so that point cannot be further addressed or changed. The authors did make a comparison between a reaction-to-observer score from the literature and the results of their models. For most of the species, the authors did find that the reaction-to-observer score from the literature did correspond with the observed fish behaviors towards the observers. Because this can't be addressed further, I feel it there isn't much more that can be added to text.

The following are specific comments that require further attention:

Not determining the exact length of transects may have introduced additional variability but this variability isn't measured by the authors or quantified. They suggest it can be assumed to be stochastic and limited. Can it be measured or is assuming it to be stochastic and limited adequate? It may or may not be adequate but a better justification should be given for this point and others that were questioned by the reviewers.

Answer: It is not possible to quantify the additional variability that is introduced by not exactly knowing how long the transects are, but we provided more information to support our assumptions: L141-143: “Although the inability to provide exact distances will introduce some additional variability that cannot be quantified, we assumed this variability to be stochastic and limited as observers were specifically trained to maintain a constant swimming speed.”

Reviewer 3 response:

The authors are not able to determine the exact lengths of the transects or quantify if additional variability was introduced because of it. They do provide more of an argument suggesting that the variability is likely limited and stochastic because each observer was trained to maintain a constant swimming speed. The new information added to the text does improve the argument to support the assumptions. Because this can't be addressed further, I feel it there isn't much more that can be added to text.

As reviewer 2 pointed out, the Introduction should be written to emphasize the importance of this study in comparison to others and its contribution to the existing literature. I don't believe that the authors response to address this comment is adequate. The foundation for the methods to be use should be established in the Introduction along with clearly stated objectives.

Answer: We have rewritten the last part of the introduction. Now the aim of the study is written down more explicitly, more early on in the manuscript and within the context of existing gaps in literature. L46-55: “Although studies comparing different video and other visual census techniques are becoming more frequent \\cite{Andradi-Brown2016,Langlois2010,Tessier2013,Watson2005,Watson2010,Wilson2018}, few have focused on the methodological aspects of video transects, such as observer bias (Table \\ref{tab:gloss}), design errors (Table \\ref{tab:gloss}), sampling variability, counting metrics, data types and data transformations \\cite{Bernard2013,Goetze2019}. Video transects are often considered a simple extension of visual census transects, although the ability to store and standardize video observations (see \\nameref{ap:add:applic}) justifies a specific methodological assessment of the technique. The aim of this study was therefore to provide a detailed assessment of the aforementioned methodological aspects and overall suitability of video transects to study reef fish assemblages.”

Reviewer 3 response:

The authors have rewritten the end of the Introduction to more clearly address the aims of the study. They didn't do an elaborate comparison with other studies but they did emphasize how their study is different than others and included a number of citations referencing studies that compare different video and visual census techniques. The authors also give a clearer explanation regarding the aim of their study later which does improve the Introduction.

Both reviewers suggested making the objectives more clear in the Introduction section. The authors mentioned that the objectives are given from line 107 to 135. However, the Introduction ends with line 78. There are 3 questions listed at the end of this section but those are not objectives. If the objectives are in fact given at the end, I cannot find them. The study objectives should be stated clearly at the end of the Introduction which they are not. I also agree with reviewers that the Introduction needs to include the a description of the type of data that is important for ecologists and managers and how the study will address ways to collect and analyze the desired types of data.

Answer: We have written down the objectives as research questions. We now specify the general aim of the study and break it down in three research questions, which are introduced more clearly now in the introduction.

Reviewer 3 response:

The objectives being written as research questions is fine, however change line 80 to "The following research questions....variables of interest:"

The choice of the data transformation was stated by the authors as being highly dependent on the research question(s) of the study. Though readers may be interested in how different transformations may affect the results of analyses, the authors didn't really address how the data transformation addresses their questions or in other words, what the point of using different data transformation was for the conclusions of the current study. There should be more to the answer than just stating that researchers might be interested in. State the importance of determining this result to researchers in the Introduction.

Answer: We included a more detailed description of the meaning/interpretation of data transformations: L274-279: ‘In addition, the considered transformation of abundance data to presence-absence data entails a broader discussion regarding the way data should be collected (i.e. identification and counting versus only identification respectively).’ And L591-595: ‘Although neglecting the abundance of a species scored worst in terms of goodness-of-fit, the difference between MinCount and presence-absence data was rather small. Therefore, in this specific case, if one wanted to reduce the time required for video analysis, the presence-absence instead of the abundance data could have been used, without much information being lost.’ We do think however that the issue of data transformation does not fit well in the introduction as it would negatively affect the structure. If we would discuss all methodological aspects in the introduction in detail it would negatively affect the clarity of the introduction. We do, however, now also mention it explicitly in the aim of the study in the introduction.

Reviewer 3 response:

Though not in the Introduction, the addition to the text regarding the meaning/interpretation of the data transformations is an improvement over the previous revision. The authors do give legitimate arguments as to why discussing these methods in the Introduction would negatively affect its structure.

The choice of statistics does make the manuscript rather complicated but justify why you are going to use these methods this in the Introduction. I understand what the authors were trying to do but it’s not explained completely within the manuscript so that the readers will understand what the objectives were and why specific analyses were done.

Answer: Discussing this in the introduction in detail would make the introduction very hard to follow and draw the attention away from the other aspects that need to be addressed in the introduction. We added following in the introduction to highlight the need for the specific methods and also refer the reader to the part of the manuscript at which more information is provided: L78-82: “The innovative, yet complex design of this study can provide new insights in the methodology of observation techniques, yet requires multiple advanced statistical methods. An overview of these methods and reasoning behind their use is provided in section \\ref{Data_analysis}.”

Reviewer 3 response:

Despite not including a discussion of the statistical methods to be used in the Introduction, the authors do give reference to the section "Data analysis" of the manuscript where the analyses are discussed.

Lastly, the following two comments given by reviewer 2 were not adequately addressed.

1. The study considers abundance and diversity of fish, but did not presented changes in species

composition, which is one of the most fundamental factors affected by different survey methods. I suggest including this explicitly. The authors did not mention if they will add anything to the manuscript regarding changes in species composition. If not, why ?

Answer: We made this more clear in the text by adding: L276-279: ‘Finally, following the predominant terminology used in literature, the multivariate analyses of abundance and presence/absence data will yield insights in the structure and species composition of fish assemblages, respectively.’ Hence the distinction between fish assemblage structures and species composition can be brought back to the use of abundance versus presence/absence data. The repercussions of different types of data (abundance versus presence/absence) is discussed in the discussion.

Reviewer 3 response:

The addition to the text addresses this question more clearly than in the first revision.

2. Similarly, the conclusions should focus on the contributions of your study. I would also suggest that given the extensive nesting design and interacting factors, the interpretation and concluding remarks should be stated clearly in the discussion as a table summarizing a set of recommendations for practitioners framed within the specific context of the study (i.e. ecosystem/habitat surveyed). It would be good for example to have carried out surveys in different habitats (need to specify in the manuscript if all transects were done in the same habitat) so the findings could have a greater applicability. The authors did not add the specific recommendations to the revision nor did they give any indication if they will or will not add them and why.

Answers: it was added that all transects were laid out in rocky habitats (L91-92). Specific habitat characteristics were not available, but habitats were very similar as rocky habitats were targeted for placement of the transects. Instead of adding a table we have extended the conclusions with the most important results of the manuscript. Most of the recommendations need to be discussed in the context of the aims, area, etc. which would take too much space in a table. Recommendations in a table without this context would lead to potential misinterpretation and we want to avoid this.

Reviewer 3 response:

The revised conclusions are an improvement over those from the previous revision. The authors have more adequately addressed this question.

Just a note, MaxN and MinCount are not density estimates. There is no unit of area included in their calculation. These can be considered abundance metrics but not density metrics.

Indeed. They are not being referred to as density metrics in the manuscript. All mention of fish densities have been removed and/or replaced by ‘fish abundance(s)’.

Reviewer 3 response:

This note was addressed in the revision.

7. PLOS authors have the option to publish the peer review history of their article (what does this mean?). If published, this will include your full peer review and any attached files.

Reviewer #3: No

---

## [Editor Report · Acceptance letter]

15 Jul 2022

PONE-D-21-06060R2 

Sampling errors and variability in video transects for assessment of reef fish assemblage structure and diversity 

Dear Dr. Bruneel:

I'm pleased to inform you that your manuscript has been deemed suitable for publication in PLOS ONE. Congratulations! Your manuscript is now with our production department. 

Kind regards, 

on behalf of

Dr. Antonio Medina Guerrero 

Academic Editor

PLOS ONE